# PTPN21 and Hook3 relieve KIF1C autoinhibition and activate intracellular transport

Nida Siddiqui [1,2,5], Alexander James Zwetsloot [1,3,5], Alice Bachmann[1,2], Daniel Roth [1,2], Hamdi Hussain [1,2], Jonathan Brandt [1,2], Irina Kaverina[4] & Anne Straube [1,2]

The kinesin-3 KIF1C is a fast organelle transporter implicated in the transport of dense core vesicles in neurons and the delivery of integrins to cell adhesions. Here we report the mechanisms of autoinhibition and release that control the activity of KIF1C. We show that the microtubule binding surface of KIF1C motor domain interacts with its stalk and that these autoinhibitory interactions are released upon binding of protein tyrosine phosphatase PTPN21. The FERM domain of PTPN21 stimulates dense core vesicle transport in primary hippocampal neurons and rescues integrin trafficking in KIF1C-depleted cells. In vitro, human full-length KIF1C is a processive, plus-end directed motor. Its landing rate onto microtubules increases in the presence of either PTPN21 FERM domain or the cargo adapter Hook3 that binds the same region of KIF1C tail. This autoinhibition release mechanism allows cargo-activated transport and might enable motors to participate in bidirectional cargo transport without undertaking a tug-of-war.

[1] Centre for Mechanochemical Cell Biology, University of Warwick, Coventry CV4 7AL, UK. [2] Division of Biomedical Sciences, Warwick Medical School, University of Warwick, Coventry CV4 7AL, UK. [3] MRC-DTP in Interdisciplinary Biomedical Research, Warwick Medical School, Coventry CV4 7AL, UK. [4] Department of Cell and Developmental Biology, Vanderbilt University Medical Center, Nashville 37232 TN, USA. [5] These authors contributed equally: Nida Siddiqui, Alexander James Zwetsloot. Correspondence and requests for materials should be addressed to A.S. (email: anne@mechanochemistry.org)

ntracellular transport is essential for cell polarity and function. Long-distance transport of cellular cargo is mediated by microtubule-based motors, dynein and kinesin. While dynein is the main transporter towards the minus end of microtubules, most kinesins walk towards the microtubule plus end[1]. However, many cargoes carry motors of both directionality in order to allow directional switching when they encounter a roadblock and the relative activity of the opposite directionality motors determines the net progress of the cargo towards the cell periphery (where usually most plus ends are located) or towards the cell centre (where microtubule minus ends are abundant)[1,2]. We have previously identified the kinesin-3 KIF1C as the motor responsible for the transport of α5β1-integrins. The delivery of integrins into cellular processes such as the tails of migrating cells allows the maturation of focal adhesion sites[3]. KIF1C is also required for the formation and microtubule-induced turnover of podosomes—protrusive actin-based adhesion structures—in both macrophages and vascular smooth muscle cells[4–6]. Moreover, KIF1C contributes to MHC class II antigen presentation, Golgi organisation and transport of Rab6-positive secretory vesicles[7,8]. In neurons, KIF1C transports dense-core vesicles both into axons and dendrites and appears to be the fastest human cargo transporter[9,10]. Consistently, human patients with missense mutations in KIF1C resulting in the absence of the protein suffer from spastic paraplegia and cerebellar dysfunction[11]. Mutations that result in reduced motor function also cause a form of hereditary spastic paraplegia[12].

KIF1C-dependent cargo moves bidirectionally even in highly polarised microtubule networks and depletion of KIF1C results in the reduction of transport in both directions[3], suggesting that KIF1C cooperates with dynein as had been suggested for other kinesin-3 motors[13]. To begin to unravel how KIF1C contributes to bidirectional cargo transport, we need to identify the mechanisms that switch KIF1C transport on and off. Most kinesin-3 motors are thought to be activated by a monomer–dimer switch, whereby cargo binding releases inhibitory intramolecular interactions of neck and stalk regions by facilitating the dimerisation of the neck coil and other coiled-coils

regions in the tail[14–17]. The motor thereby transitions from an inactive, diffusive monomer to a processive dimer. In the alternative tail-block model, the motors are stable dimers, but regions of the tail interact with the motor or neck domains and interfere with motor activity until cargo binding occupies the tail region and releases the motor[18–20].

Here we show that KIF1C is a stable dimer that is autoinhibited by intramolecular interactions of the stalk domain with the microtubule binding interface of the motor domain. We demonstrate that protein tyrosine phosphatase N21 (PTPN21) activates KIF1C by binding to the stalk region. This function does not require catalytic activity of the phosphatase, and its N-terminal FERM domain alone is sufficient to stimulate the transport of KIF1C cargoes in cells as well as increasing the landing rate of KIF1C on microtubules in vitro. The cargo adapter Hook3 binds KIF1C in the same region and activates KIF1C in a similar way, suggesting that both cargo binding and regulatory proteins might contribute to the directional switching of KIF1C–dynein transport complexes.

## Results

**KIF1C is an autoinhibited dimer.** To determine the mechanism of KIF1C regulation, we first aimed to determine its mode of autoinhibition. The monomer–dimer-switch and tail-block models can be distinguished by determining the oligomeric state of the motor. Thus we performed classic hydrodynamic analysis using glycerol gradients and size exclusion chromatography on both purified recombinant full-length human KIF1C-GFP from insect cells and KIF1C-Flag in human cell lysate (Fig. 1a–c, Supplementary Fig. 1). The sedimentation coefficient and Stokes radius were determined in comparison to standard proteins. We find that the apparent molecular weight determined both at physiological levels of salt (150 mM) and in high (500 mM) salt buffer is consistent with KIF1C being a dimer (KIF1C-GFP: apparent MW = 272 ± 57 kDa, expected dimer MW = 308 kDa; KIF1C-Flag: apparent MW = 178 ± 45 kDa, expected dimer MW = 251 kDa, all errors are SEM based on

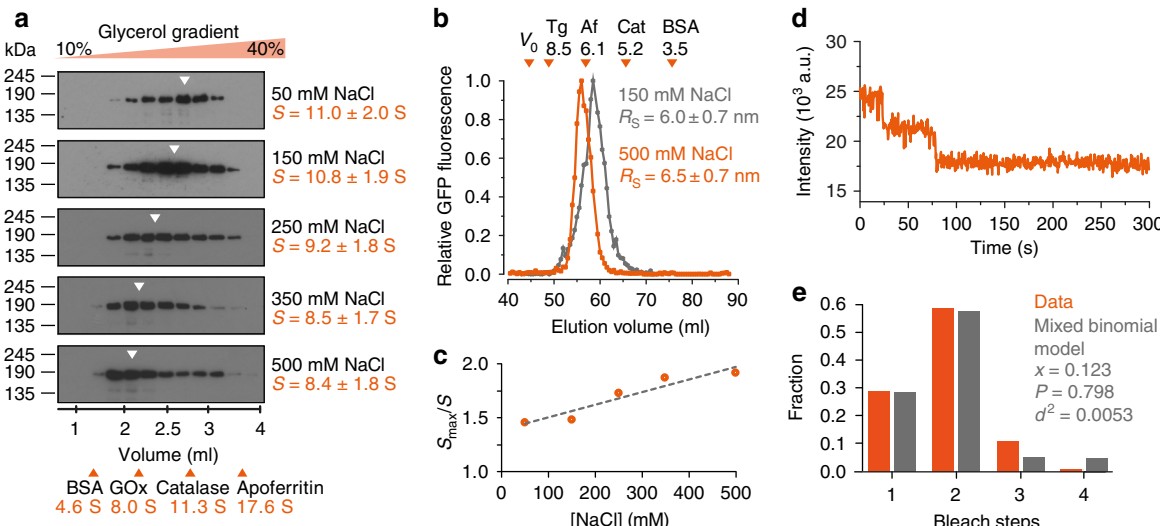

**Fig. 1** KIF1C is a dimer. **a** Fractions from glycerol gradients of KIF1C-GFP at different salt concentrations as indicated. Elution peaks of standard proteins are indicated with orange arrowheads. GOx, glucose oxidase. Errors are SEM. Uncropped gels are provided in Source Data file. **b** Size exclusion chromatography of KIF1C-GFP at 150 mM NaCl (grey) and 500 mM NaCl (orange). Elution peaks of standard proteins (Tg thyroglobulin, Af apoferritin, Cat catalase, BSA bovine serum albumin) and void volume $V_0$ are indicated by orange arrowheads. Errors are SEM. **c** Frictional coefficient of KIF1C-GFP at different salt concentrations indicating that KIF1C elongates with increasing ionic strength. **d**, **e** Bleach curve of KIF1C-GFP on microtubules showing discrete steps in fluorescent decay in **d**. Experimentally determined bleach steps are shown in **e** together with best fit to a mixed binomial model of dimers and tetramers with $x$ being the fraction of tetramer and $p$ the fraction of active GFP molecules. $n$=108 motors. Data are provided in Source Data file

errors of calibration curve fit parameters and estimated precision of peak location as 1/2 of fraction size). Consistent with this, the majority of KIF1C-GFP molecules bound to microtubules in the presence of non-hydrolysable AMPPNP show two bleach steps and the distribution of bleach steps found fits best to a simulation of 88% dimers and 12% tetramers (Fig. 1d, e), assuming that about 80% of GFPs are active, which is realistic based on previous findings[21]. Interestingly, KIF1C elongates at increasing salt concentrations from a moderately elongated conformation (frictional coefficient of 1.5) at physiological salt to highly elongated (frictional coefficient = 1.9) at 500 mM (Fig. 1c), suggesting that intramolecular electrostatic interactions might hold the KIF1C dimer in a folded, autoinhibited state.

To determine the interaction surfaces involved in a possible autoinhibited state, we performed crosslink mass spectrometry. Purified KIF1C was treated with the 11 Å crosslinker BS3 or with the zero length crosslinker EDC, digested with trypsin and then subjected to tandem mass spectrometry analysis. Crosslinked peptides were identified using StavroX[22]. Only crosslinked peptides whose identity could be verified by extensive fragmentation with none or few unexplained major peaks were retained (Supplementary Figs. 2 and 3). These high-confidence crosslinks were between K273-K591, K273-K645, K464-K645, K640-K645, K464-E606 and E644-K854 (Fig. 2a, Supplementary Figs. 2 and 3), showing that the end of the FHA domain and the third coiled-coil contact the motor domain near K273. This residue is within alpha-helix 4 at the centre of the microtubule interaction interface of the motor domain[23] (Fig. 2b). In the presence of 500 mM NaCl, the abundance of the stalk-to-motor domain crosslinks was dramatically reduced consistent with the observed elongation of KIF1C dimer in glycerol gradients (Supplementary Figs. 2e and 1c). These findings suggest that KIF1C adopts a more

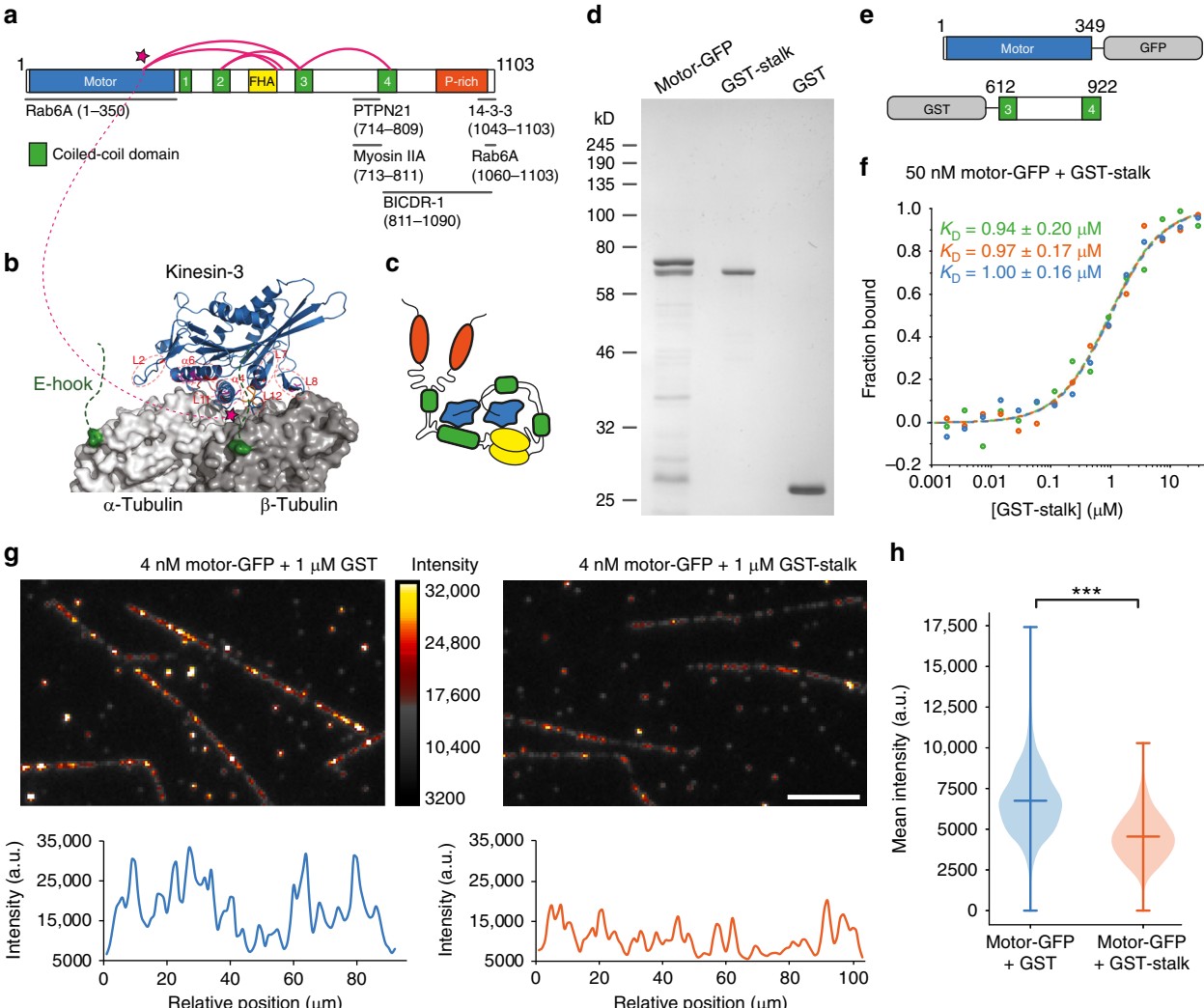

**Fig. 2** KIF1C is autoinhibited by intramolecular interactions of stalk and motor domain. **a** Schematic primary structure of KIF1C with motor domain (blue), coiled-coil domains (green), FHA domain (yellow), and Proline-rich domain (orange). Crosslinks identified using mass spectrometry after treatment with BS3 or EDC are shown as magenta loops. Published binding sites for KIF1C interactors are indicated below. See Figs. S2 and S3 for ion fragmentation of crosslinked peptides. **b** Motor domain of related KIF1A on tubulin. The region in the motor domain that interacts with KIF1C stalk is indicated by magenta stars. **c** Hypothetical model of autoinhibited KIF1C conformation based on identified crosslinks. **d** Coomassie-stained SDS-PAGE gel of purified KIF1C motor domain, stalk domain and GST control. **e** Schematic primary structure of KIF1C motor and stalk proteins used here. **f** Binding affinity measurements and Kd model fits from three microscale thermophoresis experiments probing the interaction of KIF1C motor domain with the stalk domain. **g** Representative TIRF images showing microtubule decoration of KIF1C motor domain in the presence of GST (control) and KIF1C tail domain. Scale bar 5 μm. Linescans of one of the microtubules from each field is shown below. **h** Quantification of mean intensity of KIF1C motor domain on microtubules. Distribution is shown together with mean and full extent of data. $n = 690$ and 659 microtubules, respectively, pooled from six experiments. *** indicates $p = 10^{-99}$ ($t$-test)

compact conformation in physiological conditions whereby the stalk domain blocks the motor domain from interacting with microtubules (Fig. 2c). To confirm this intramolecular interaction and the autoinhibition directly, we purified bacterially expressed KIF1C motor domain and KIF1C stalk domains (Fig. 2d, e). Using microscale thermophoresis, we measured the affinity of motor and stalk domains as 1 μM (Fig. 2f). Consistent with the idea that the stalk competes with microtubule binding, we found that microtubule decoration by the KIF1C motor domain was significantly reduced in the presence of 1 μM stalk domain, but not when 1 μM GST was added as control (Fig. 2g, h). Thus KIF1C is a stable dimer that blocks its microtubule-binding interface with an intramolecular interaction in its autoinhibited conformation.

**PTPN21 stimulates transport of KIF1C cargoes**. We next aimed to identify a regulator that could engage with the stalk domain and thereby activate the motor. Two proteins had previously been shown to interact with the KIF1C stalk domain, non-muscle Myosin IIA[4] and protein tyrosine phosphatase N21 (PTPN21, also known as PTPD1)[24] (Fig. 2a). We reasoned that a KIF1C activator would be required for KIF1C function in cells and thus would phenocopy KIF1C depletion. As a readout of diminished KIF1C activity, we used the reduced podosome number in vascular smooth muscle cells that we reported previously[5]. Myosin IIA inhibition did not phenocopy KIF1C depletion. Instead, inhibition of Myosin IIA with Blebbistatin or indirectly via inhibition of Rho kinase using Y27632 did not result in reduced podosome number, while remaining actin stress fibres were efficiently removed (Fig. 3a–d). As depletion of Myosin IIA did also have opposite effects on the stability of cell tails and directional persistence of cell migration we previously reported for KIF1C depletion[3], we excluded Myosin IIA as being involved in KIF1C activation. In contrast, PTPN21 depletion using siRNA resulted in a dramatic reduction of podosome number (Fig. 3e–h). To confirm specificity of the RNAi we rescued the phenotype with HA-tagged PTPN21. Interestingly, the catalytically inactive mutant PTPN21$_{C1108S}$ (ref. [25]) could also fully rescue the PTPN21 depletion phenotype (Fig. 3e, f), suggesting that a scaffold function is required rather than phosphatase activity.

Strikingly, expression of PTPN21$_{C1108S}$ also efficiently compensated for partial depletion of KIF1C and fully rescued podosome formation (Fig. 4a–c). An N-terminal 378 amino acid fragment of PTPN21, which contains a FERM domain, was previously shown to be sufficient to interact with KIF1C[24]. Therefore we tested this construct, and found that the PTPN21 FERM domain alone was sufficient to rescue the KIF1C depletion phenotype (Fig. 4a–c). This suggests that PTPN21 scaffolding could activate the remaining pool of KIF1C, which is reduced to about 25–30% in cells treated with siKIF1C-2 compared to control cells[3,5]. Alternatively, PTPN21 could also activate another kinesin to compensate for KIF1C depletion. KIF16B, another kinesin-3, was an obvious candidate for this as its interaction with PTPN21 FERM domain had been shown already[26]. To discriminate between these two possibilities, we depleted KIF1C and KIF16B individually and simultaneously in PTPN21 FERM rescue experiments. KIF16B depletion on its own resulted only in a mild reduction in podosome numbers and expression of FERM domain significantly increased the number of podosomes formed in control cells and in those where either of the kinesins was depleted. However, PTPN21 FERM did not increase podosome numbers in cells that were depleted for both KIF1C and KIF16B (Fig. 4d, Supplementary Fig. 4). These findings suggest that PTPN21 rescue depends on the presence of a sizeable pool of either KIF1C or KIF16B. To explore this activity further, we

investigated more directly whether PTPN21 stimulates KIF1C-dependent transport processes. We have shown previously that KIF1C is required for the bidirectional transport of integrin-containing vesicles in migrating RPE1 cells[3]. The depletion of KIF1C results in a reduction of both plus-end and minus-end-directed transport and an increase in stationary vesicles. Expression of either wild-type PTPN21, the catalytically inactive PTPN21$_{C1108S}$ mutant or PTPN21$_{FERM}$ reactivated integrin vesicle transport in KIF1C-depleted cells (Fig. 4e, f). To test whether the ability of PTPN21$_{FERM}$ to activate KIF1C-dependent transport universally applies, we next isolated primary hippocampal neurons and observed the transport of dense-core vesicles labelled with NPY-RFP in the presence of either a control plasmid or pFERM. Consistent with a function to activate KIF1C-dependent transport, we observed a dramatically increased frequency of vesicles moving in anterograde direction through the neurites in cells expressing the FERM domain of PTPN21 (Fig. 4g–i).

**PTPN21 relieves KIF1C autoinhibition**. Our experiments in cells identified PTPN21 as a potential activator of KIF1C. To confirm that the PTPN21 FERM domain can directly activate KIF1C, and to further interrogate the mechanism of activation, we reconstituted KIF1C motility in vitro. Hilyte647- and biotin-labelled microtubules were assembled in the presence of GTP, then stabilised with Taxol and attached to a PLL-PEG-biotin-coated coverslip. Six hundred picomolar KIF1C was added in the presence of 1 mM ATP and an ATP-regenerating system. At 25 °C, individual KIF1C-GFP dimers moved processively towards the plus end of the microtubule where they accumulated (Fig. 5a). The average speed and run length were 0.45 μm/s and 8.6 μm respectively (Fig. 5d). We then purified recombinant PTPN21$_{FERM}$-6His and for comparison Ezrin$_{FERM}$-6His from *Escherichia coli* (Fig. 5c). Consistent with being an activator, the landing rate of KIF1C motors increased by about 40% in the presence of PTPN21$_{FERM}$ (Fig. 5b–d). The frequency of observing running motors was also increased by 40%. Ezrin$_{FERM}$, acting as a negative control, did not significantly affect the landing rate or the frequency of running motors (Fig. 5b–d). These findings are consistent with the idea that PTPN21 opens the KIF1C motor by binding to its tail domain and thereby relieves autoinhibition and increases the binding rate of the motor.

If PTPN21 relieves the autoinhibition, the following two predictions should hold true: (i) PTPN21$_{FERM}$ binding should remove the interactions between stalk and motor domain of KIF1C and (ii) deleting the region required for autoinhibition should result in a dominant-active motor that cannot be further activated by binding of PTPN21$_{FERM}$. To test the first prediction, we performed crosslinking mass spectrometry experiments with KIF1C and PTPN21$_{FERM}$. We found two new crosslinks, but could no longer find the intramolecular crosslinks in KIF1C (Fig. 5e, f). The new crosslinks show interactions of the FERM domain with lysines 591 and 645 in KIF1C (Supplementary Fig. 5). Because the crosslinks between motor domain and stalk were absent in the presence of PTPN21$_{FERM}$, the relative abundance of the un-crosslinked LKEGANINK peptide at the microtubule binding interface of KIF1C increased more than twofold in the KIF1C + PTPN21$_{FERM}$ sample compared to KIF1C alone (Fig. 5e). These data confirm that the presence of the FERM domain is capable of freeing the motor domain, and relieving the autoinhibition of KIF1C. To test the second prediction, we generated two deletion mutants, KIF1CΔCC3 in which we removed coiled-coil 3, which makes the predominant contact with the motor domain in the autoinhibited conformation and KIF1CΔS where we removed coiled-coil 3 and the stalk region

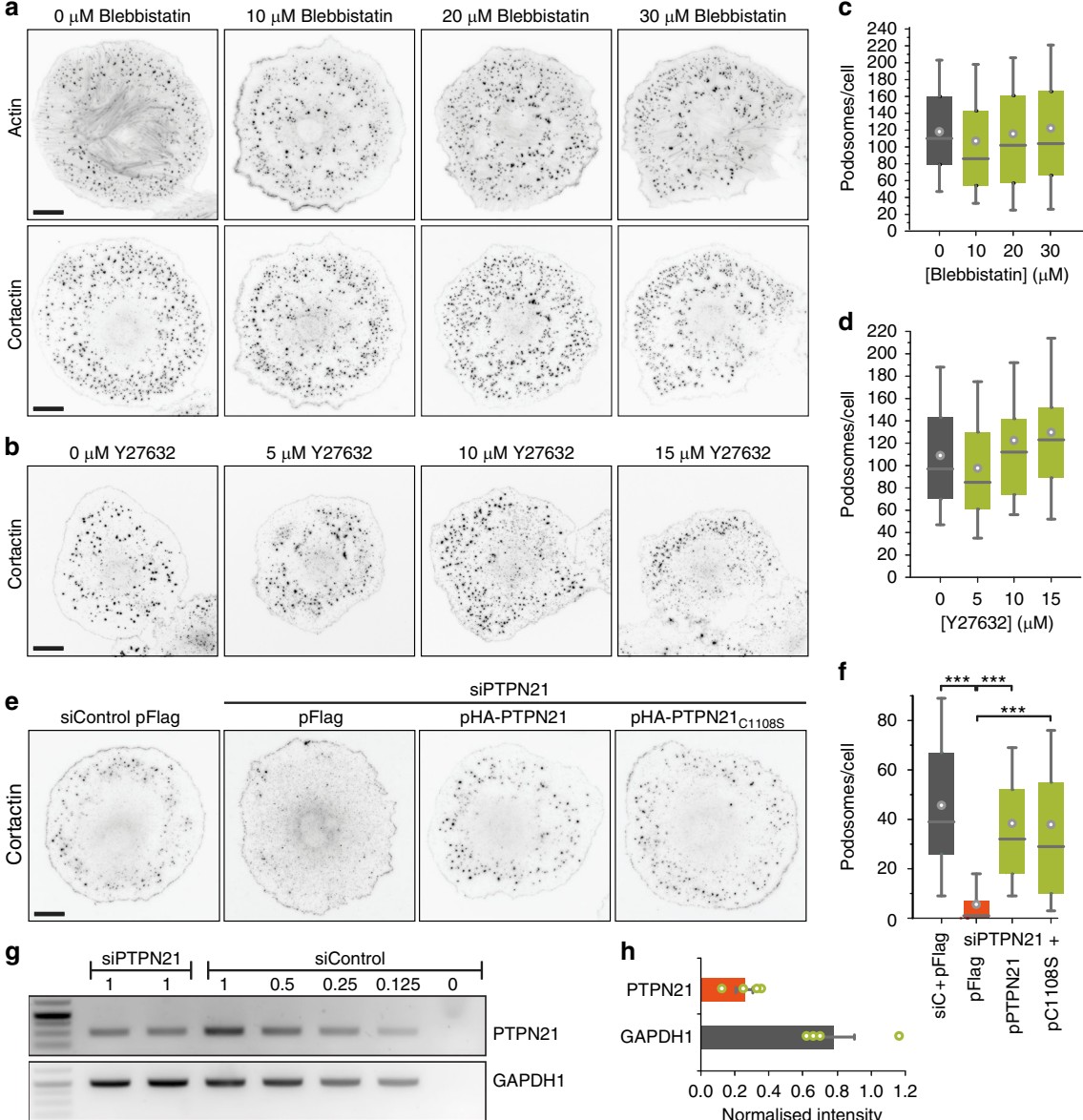

**Fig. 3** PTPN21, but not Myosin IIA, is required for podosome formation. **a**, **b** A7r5 cells treated with 5 μM PDBu and different concentrations of Blebbistatin or Y27632 for 1 h before staining for actin and cortactin. Scale bars 20 μm. **e** A7r5 cells transfected with siControl and siPTPN21 and either control plasmid (pFlag) or an RNAi-protected construct of wild-type PTPN21 or a catalytically inactive mutant PTPN21C1108S were treated with 5 μM PDBu for 1 h. Scale bar 20 μm. **c**, **d**, **f** Quantification of podosome numbers under different experimental conditions as indicated. $n = 90$ cells pooled from three independent experiments. Box plots show quartiles with 10/90% whiskers and mean indicated by a circle. Statistical significance with $p < 0.05$ is indicated with asterisks, *** represents $p < 0.0005$ (Mann–Whitney $U$-test with Bonferroni). Data are provided in Source Data file. **g**, **h** RT-PCR from random-primed cDNA of A7r5 cells treated with siRNA as indicated. For each experiment, duplicates of siPTPN21 and five different concentrations of cDNA from siControl were analysed. Quantification of RT-PCR band intensities for siPTPN21 relative to siControl standard curve from four independent experiments is shown as mean ± SEM with overlaid data points. Uncropped gels and data are provided in Source Data file

previously identified as minimal PTPN21-binding domain (Fig. 6a). First, we assessed whether either or both constructs are hyperactive by localisation of the construct in cells. We had described previously that KIF1C accumulates specifically in tails of migrating RPE1 cells[3]. Expression of KIF1CΔCC3-GFP and KIF1CΔS-GFP in RPE1 cells resulted in accumulation of the construct in cell tails and also elsewhere at the periphery of the cell (Fig. 6b). As tail accumulation is sensitive to tail state, i.e. whether it is forming or retracting[3], we used KIF1C-mCherry as an internal control. In comparison to KIF1C-GFP, which was enriched in tails to a similar extent as KIF1C-mCherry (tail: cytoplasm ratio 6.1 ± 1.3 for KIF1C-mCherry and 7.4 ± 1.2 for

KIF1C-GFP, $n = 26$ cells, errors are SEM), KIF1CΔCC3-GFP was almost threefold more enriched (tail:cytoplasm ratio 3.9 ± 0.6 for KIF1C-mCherry and 13.6 ± 2.1 for KIF1CΔCC3-GFP, $n = 32$ cells), and KIF1CΔS-GFP was fourfold more enriched (tail: cytoplasm ratio 1.8 ± 0.3 for KIF1C-mCherry and 7.7 ± 0.8 for KIF1CΔS-GFP, $n = 39$ cells) (Fig. 6c). Secondly, we purified recombinant KIF1CΔCC3-GFP and KIF1CΔS-GFP from insect cells (Fig. 6d) and performed single-molecule assays. KIF1CΔCC3-GFP had a threefold higher landing rate compared to wild-type KIF1C-GFP, and addition of PTPN21_FERM did not further increase the landing rate of KIF1CΔCC3-GFP (Fig. 6e–g, k). KIF1CΔS-GFP had a 20-fold higher landing rate

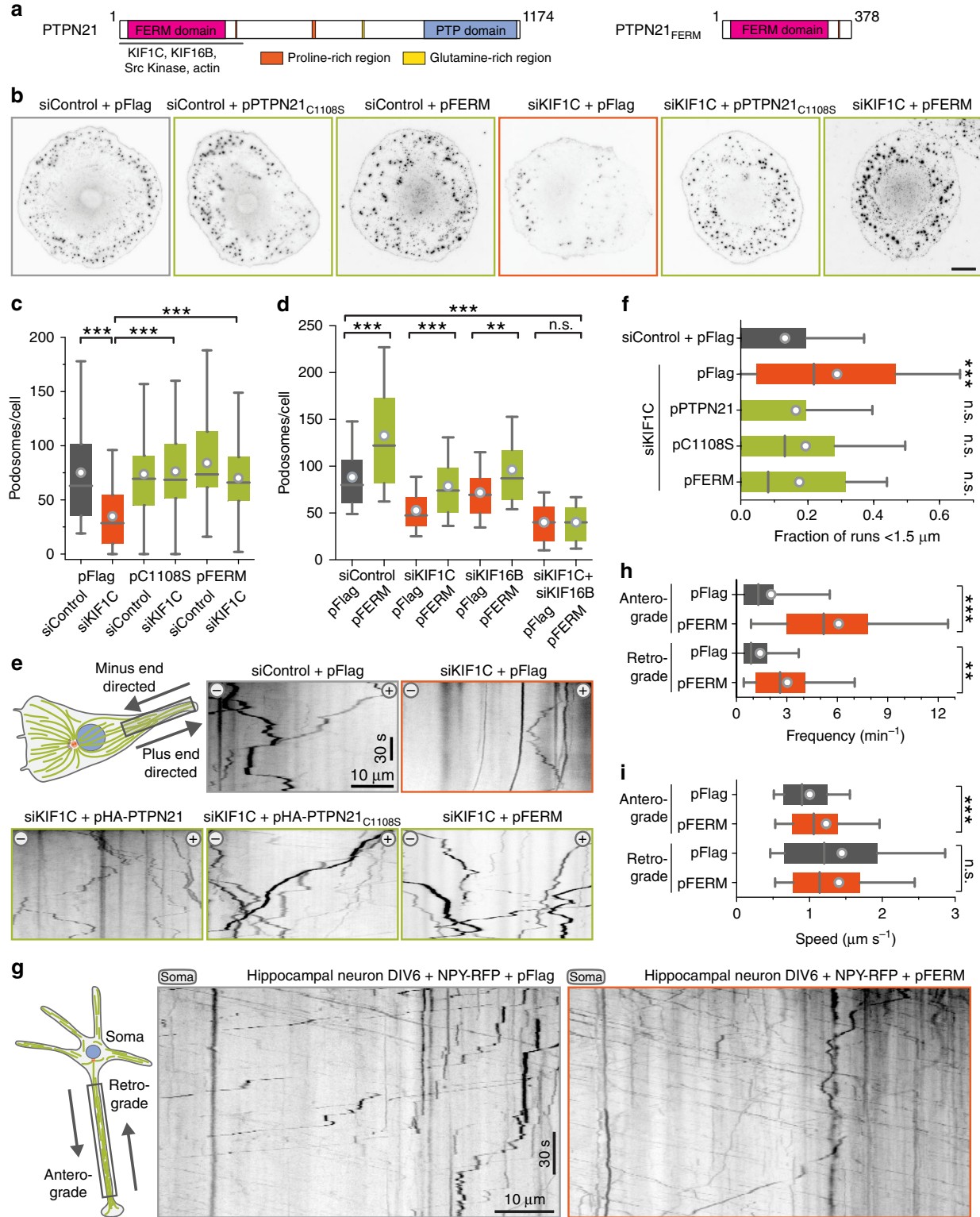

than KIF1C-GFP and also moved at about threefold higher average speed (Fig. 6h–m). Therefore, these experiments confirm that binding of PTPN21 to the stalk domain of KIF1C relieves the autoinhibition of KIF1C and thereby enables the motor to engage with microtubules.

**Hook3 binds to KIF1C stalk and also activates KIF1C.** Finally, we wondered how universal this mechanism is and whether other

proteins binding KIF1C stalk could activate KIF1C in a similar manner. To identify KIF1C stalk interactors we performed a BioID experiment with full-length KIF1C and KIF1CΔS (Fig. 7a). We identified 43 proteins or clusters of proteins isolated with streptavidin-beads upon biotinylation that were significantly enriched in either the KIF1C-BioID2 or KIF1CΔS-BioID2 samples. Eight of those were enriched more than 10-fold in the full-length sample. The most significant hit was Hook3 (Fig. 7b) which was 50-fold enriched in the full-length sample and thus

**Fig. 4** PTPN21 activates intracellular transport. **a** Primary structure of PTPN21 and N-terminal fragment used in this study. The region identified to interact with kinesins, actin and Src kinase is indicated below. **b**, **c** A7r5 cells treated with 5 μM PDBu for 1 h and stained for cortactin as a marker for podosomes. Expression of catalytically inactive PTPN21 or just the FERM domain rescues the podosome formation phenotype of KIF1C-depleted cells. $n = 60$ cells. ***$p < 0.0001$ (Mann–Whitney U-test with Bonferroni). Scale bar 20 μm. Data are provided in Source Data file. **d** Podosome formation in A7r5 cells treated with siKIF1C, siKIF16B or both. Expression of the FERM domain increases podosome formation unless both kinesins are depleted. $n = 90$ cells pooled from three independent experiments. ***$p < 0.0001$ **$p < 0.001$ (Mann–Whitney U-test with Bonferroni). See Supplementary Fig. 4 for representative examples. Data are provided in Source Data file. **e**, **f** Kymographs of α5-integrin vesicles in the tail of migrating RPE1 cells. KIF1C depletion results in increase of vesicles moving less than 1.5 μm. Expression of various PTPN21 constructs suppresses the KIF1C depletion phenotype. $n = 36–158$ cells pooled from 3 to 11 independent experiments. ***$p < 0.0001$ and n.s. $p > 0.5$ (ANOVA + Tukey post hoc relative to siControl). Data are provided in Source Data file. **g–i** Representative kymographs show primary hippocampal neurons isolated from a P2 mouse, transfected with NPY-RFP and either pFlag (as control) or pFERM. Number of NPY-positive vesicles passing a location per minute and the average speed of vesicles is shown for anterograde and retrograde movement. Data pooled from three independent experiments (DIV5-6). $n = 3760$ neurites/197,953,136,480 vesicles. ***$p < 0.0001$; **$p < 0.005$; n.s. $p > 0.5$ (Kolmogorov–Smirnov)

appeared as a possible KIF1C stalk interactor. We confirmed that both Hook3 and PTPN21 bind to KIF1C in a stalk-dependent manner using co-immunoprecipitation from HEK293 cells (Fig. 7c, Supplementary Fig. 6). Hook3 has previously been identified as an activator of dynein/dynactin—an activity requiring its N-terminal globular domain and all three coiled-coil domains[27–30]. In contrast, the C-terminus of Hook3 was shown to interact with KIF1C[31]. To test whether Hook3 could also activate KIF1C, we purified recombinant human Hook3 with a C-terminal SNAP-tag from insect cells (Fig. 7d), labelled it with Alexa647 and performed single-molecule assays with KIF1C. In the presence of Hook3, the landing rate of KIF1C increased twofold (Fig. 7e, f), thus confirming that Hook3 functions as an activator similarly to PTPN21 by binding to the KIF1C stalk region adjacent to the site engaged in intramolecular interactions of KIF1C during autoinhibition. As we could observe that Hook3 was co-transported with KIF1C (Fig. 7e), we asked whether PTPN21 also maintained contact with KIF1C after activating it. We purified and labelled FERM-SNAPf with Alexa647 similarly to our Hook3 construct and performed two-colour TIRF assays both with KIF1C-GFP and KIF1CΔS-GFP. Both activators were co-transported with full-length KIF1C, but not with KIF1CΔS (Fig. 8a–d). This suggests that both PTPN21 and Hook3 act as scaffolds to activate KIF1C by engaging with its stalk region (Fig. 8e, f).

## Discussion

Our findings support a model whereby KIF1C is a stable dimer that is held in an autoinhibited conformation by interaction of its stalk region including the third coiled-coil domain with the microtubule binding surface of the motor domain (Fig. 8e). Autoinhibition is relieved upon binding of PTPN21 FERM domain or Hook3, allowing the motor domain to engage with microtubules (Fig. 8f). While the model agrees with the finding that KIF1C motors are dimeric in cells[32], this is in contrast to the mode of autoinhibition described for other kinesin-3 motors, KIF1A, Unc104, KIF16B, KIF13A and KIF13B that all undergo a monomer–dimer transition[14–17]. However, interactions of the stalk or tail region with the motor domain have also been described for KIF13B and KIF16B[19,20]. These might act as a second layer of activity control once dimers are formed or stabilise the inhibited monomer conformation. Our data suggest that KIF1C autoinhibition acts by steric blockage of the microtubule binding. In contrast, kinesin-1, which is also autoregulated by a tail-block mechanism, is inhibited by crosslinking of the motor domains that prevent movement required for neck linker undocking[33,34].

We identify the phosphatase PTPN21 as a KIF1C activator. The first 378 amino acids containing the FERM domain are sufficient to activate KIF1C in vitro and in cells and does neither require the catalytic activity nor the phosphatase domain of PTPN21. Even though a scaffold activity is sufficient, it is still possible that PTPN21-mediated dephosphorylation of KIF1C[24] can further modulate the activity of KIF1C, modify KIF1C cargo, cargo adapters or the activity of adjacent motors, a possibility that will be interesting to explore in a future study. PTPN21 has been shown to dynamically localise to focal adhesions[35] or EGF receptor recycling sites[36]. It is possible that KIF1C-mediated transport facilitates the efficient turnover of PTPN21 at these sites. Thus the phosphatase could be both a cargo and a regulator of KIF1C. We have implicated KIF1C previously in the transport of integrins required for the maturation of trailing focal adhesions[3]. It is possible that additionally, KIF1C transport facilitates PTPN21-mediated regulation of Src tyrosine kinase and FAK signalling that promote cell adhesion and migration[35].

PTPN21 FERM was able to compensate for depletion of KIF1C by activating KIF16B, a highly processive early endosome transporter that has been shown to also interact with PTPN21 FERM[15,26,37]. The expression of the FERM domain efficiently stimulated dense-core vesicle transport in primary neurons. PTPN21 has been associated with schizophrenia in a genome-wide association study[38] and shown to promote neuronal survival and growth[39,40]. While the latter is thought to occur via NRG3 or ERK1/2 signalling, neuronal function and morphology is compromised when microtubule transport is perturbed and the function of PTPN21 as neuronal transport regulator might contribute to the complications in schizophrenia patients with PTPN21 mutations. Further, cAMP/PKA pathway stimulation of kinesin-1 has been shown to reverse aging defects in *Drosophila* neurons, and a similar age-driven decrease in KIF1C transport of dense-core vesicles and other organelles may have similar effects in neurodegenerative diseases[41].

The findings that Hook3 can activate both dynein/dynactin[27,29,30] and KIF1C (this study), and that the binding sites for these opposite directionality motors are non-overlapping[29,31], suggests that Hook3 could simultaneously bind to KIF1C and dynein/dynactin and provide a scaffold for bidirectional cargo transport. Evidence for the existence of a complex of dynein/dynactin, KIF1C and Hook3 has recently been provided in a preprinted manuscript[42]. We note that this study did not report an activation of KIF1C upon binding of Hook3; however, this is based solely on the analysis of speed and run lengths, while we find that activation primarily increases KIF1C landing rates. How the directional switching would be orchestrated in such a KIF1C-DDH complex is an exciting question for the future. It is important to note that Hook3 is not the only dynein cargo adapter which binds KIF1C. BICDR1 has been shown to bind to the proline-rich C-terminal region of KIF1C[9], and BICD2

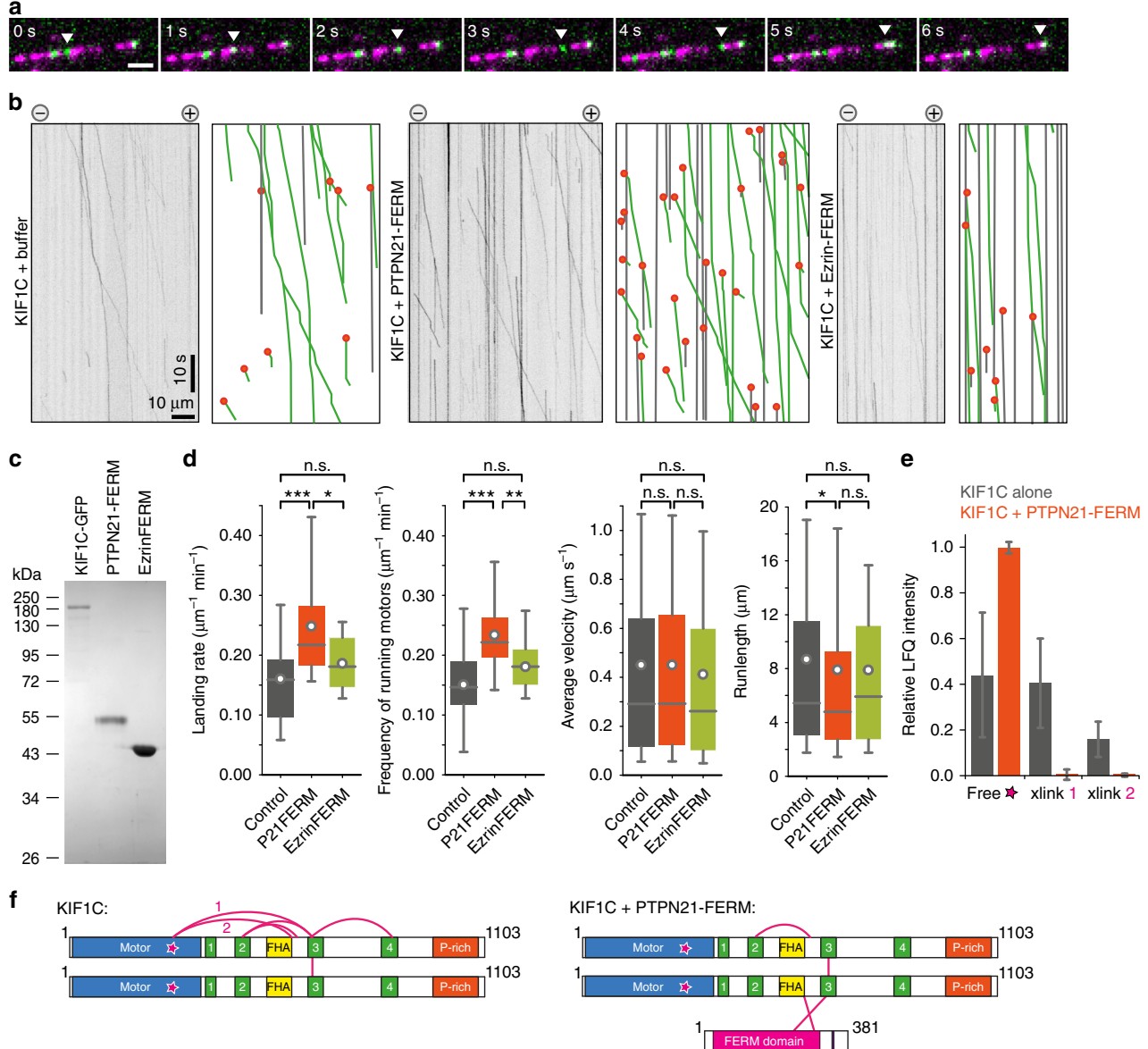

**Fig. 5** PTPN21 FERM domain activates KIF1C in vitro. **a** KIF1C-GFP (green) is a processive motor in single-molecule assays on Taxol-stabilised microtubules (magenta). Scale bar 2 μm. **b** Representative kymographs from single-molecule experiments of KIF1C in the presence of FERM domains of PTPN21 and Ezrin. Grey lines indicate immobile motors; green lines running motors and orange dots landing events. **c** Coomassie-stained SDS-PAGE of purified KIF1C-GFP and FERM domains of PTPN21 and Ezrin. **d** Quantification of landing rate, frequency of running motors (>25 nm/s), average velocity and run length. $n = 30$, 39 and 21 microtubules respectively from three independent experiments. $*p < 0.05$, $**p < 0.005$, $***p < 0.0005$, n.s. $p > 0.05$ (t-test with Bonferroni correction). Data are provided in Source Data file. **e, f** Crosslinking mass spectrometry of KIF1C in the presence of PTPN21-FERM identified two crosslinks between KIF1C and the FERM domain. Crosslinks in KIF1C alone are shown for comparison. Star indicates peptide in motor domain that is found in crosslinks with stalk domain (labelled 1 and 2). Relative abundance of single peptide LKEGANINK (star) and the two crosslinked peptides between motor and stalk domain are shown for KIF1C alone and KIF1C + PTPN21-FERM. Data show mean ± SD for three independent experiments. See Supplementary Figs. 2, 3 and 5 for ion fragmentation spectra. Data are provided in Source Data file

appears to interact with KIF1C biochemically[43]. Whether BICDR1 or BICD2 are able to activate the motor is unclear, but it is possible that different adapters not only mediate linkage to a different set of cargoes, but also recruit opposite polarity motors in different conformations and thus relative activity. For dynein/dynactin, such a difference is seen in BICD2 recruiting only one pair of dynein heavy chains while BICDR1 and Hook3 recruit two pairs and thus are able to exert higher forces[28]. BICDR1 also binds Rab6 and recruits both dynein/dynactin and KIF1C to participate in the transport of secretory vesicles[9]. Rab6 in turn has been shown to bind and inhibit the KIF1C motor domain[7].

This could provide a potential mechanism for a second layer of regulatory control of KIF1C activity to facilitate its minus end-directed transport with dynein-dynactin-Hook3.

Taken together, we provide mechanistic insight into the regulation of KIF1C, a fast long-distance neuronal transporter. We show that KIF1C is activated by a scaffold function of PTPN21 and the dynein cargo adapter Hook3, but the mechanism of autoinhibition release described here is likely to be universal and we expect cargoes and further scaffold proteins binding to the stalk region around the third coiled-coil in KIF1C to also activate the motor and initiate transport along microtubules. This opens

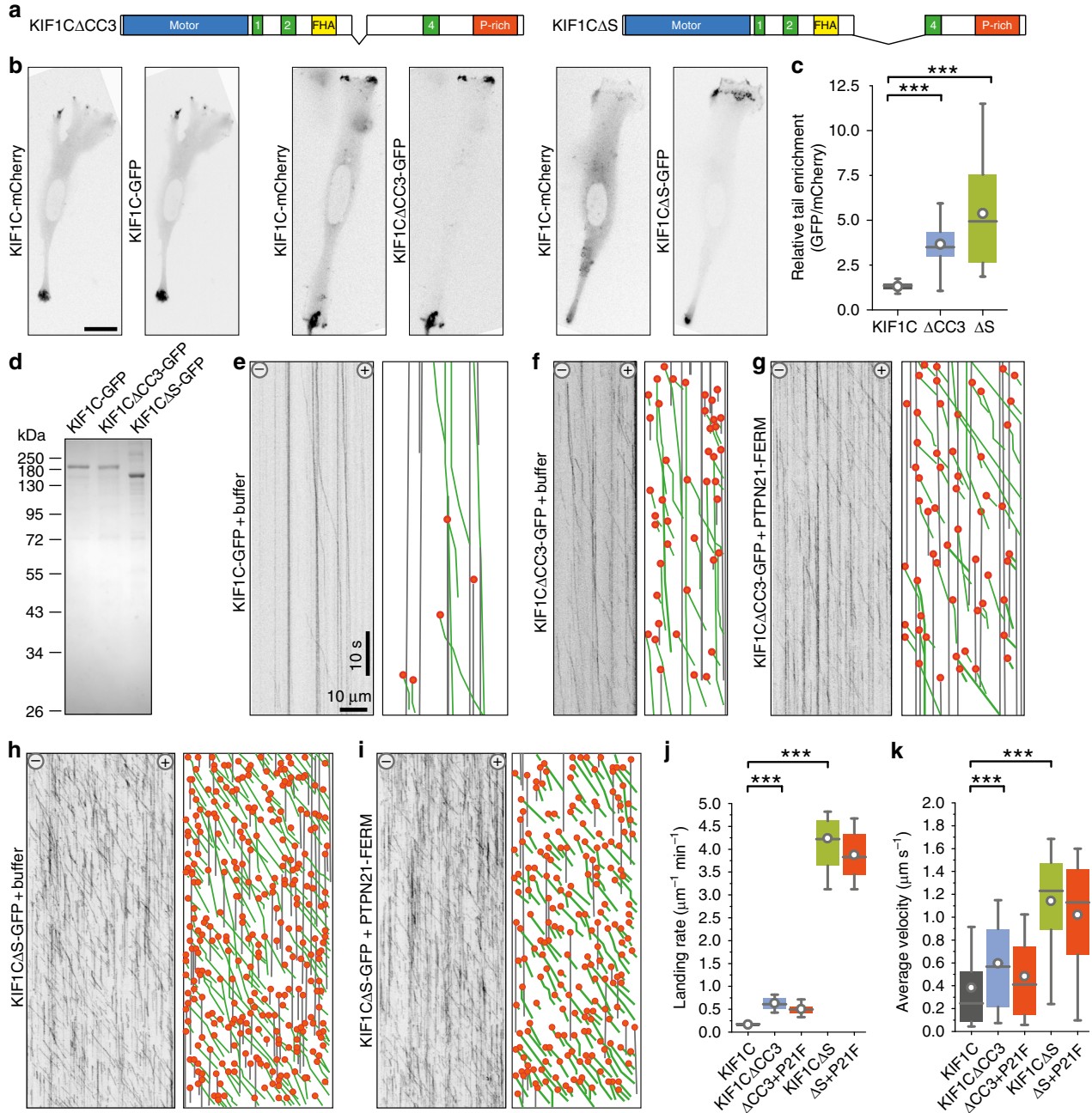

**Fig. 6** Deletion of KIF1C stalk results in a hyperactive motor. **a** Primary structure of hyperactive KIF1C deletion mutants used here. **b** Representative RPE1 cells co-transfected with full-length KIF1C-mCherry and either KIF1C-GFP, KIF1CΔCC3-GFP or KIF1CΔS-GFP and imaged 36 h post transfection. Scale bar is 20 μm. **c** Box plot shows ratio of KIF1C enrichment at the tail relative to cytoplasmic levels for GFP vs. mCherry channel. ***$p < 0.0005$ (Kolmogorov–Smirnov). $n = 26, 32, 39$ cells. Data are provided in Source Data file. **d** Coomassie-stained SDS-PAGE of purified KIF1C-GFP and deletion mutants. **e–i** Representative kymographs from single-molecule experiments. Grey lines indicate immobile motors, green lines running motors and orange dots landing events. **j, k** Frequency of KIF1C landing events and average velocity of running motors (>25 nm/s). $n = 17, 82, 58, 22, 24$ microtubules, respectively, pooled from three experiments. ***$p < 0.0005$ ($t$-test with Bonferroni correction). Data are provided in Source Data file

up new research avenues into how KIF1C activity is controlled in space and time in cells.

## Methods

**Plasmids and siRNAs.** The following plasmids used in this study were described previously: pKIF1C[RIP2]GFP (ref. [5]), pKIF1C-mCherry, pKIF1C-2xFlag, pFlag[3], pα5-integrin-GFP[44], pNPY-RFP[45], pHA-PTPN21-WT and pHA-PTPN21-C1108S (ref. [25]).

pFastBac-M13-6His-KIF1CGFP was generated by digesting pKIF1C[RIP2]GFP with EcoRI–MfeI and replacing it in the backbone of pFastBac-M13 (Invitrogen). A resulting frameshift between the 6His tag and the N-terminus of KIF1C was corrected by cutting with EcoRI, mung bean nuclease treatment and religation of the vector.

KIF1CΔCC3 (i.e. KIF1CΔ623–679) and KIF1CΔS (i.e. KIF1CΔ623–825) deletions were generated by PCR, introducing SalI restriction sites after amino acid position D679 using oligo AS370 and after amino acid position D825 using AS371 with AS83 as the reverse primer (see Supplementary Table 1 for sequences of DNA oligonucleotides used in this study). The SalI–BamHI fragment in

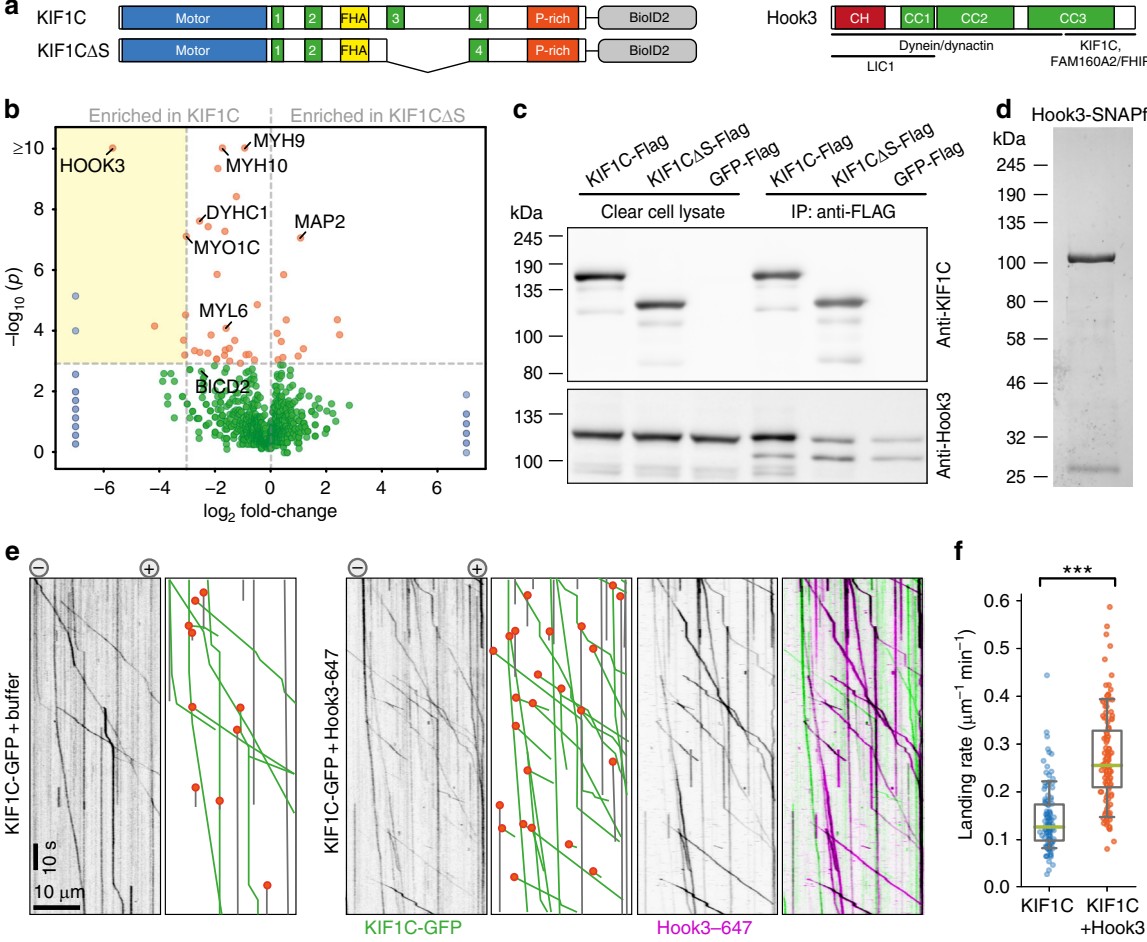

**Fig. 7** Hook3 activates KIF1C. **a** Schematic primary structure of constructs used for BioID: full-length KIF1C and KIF1CΔS. Hook3 is also shown together with regions of confirmed interactors. **b** Volcano plot shows ratio of proteins enriched in KIF1CΔS relative to full-length KIF1C relative to p-value of Fisher's exact test. Proteins in blue were only present in one of the samples. Proteins labelled in orange are significantly enriched across the repeats. The yellow box includes all significant hits that are at least 10-fold enriched in full-length KIF1C sample. n = 3 independent experiments, one of which included three replicates. See Supplementary Data 1. **c** Co-immunoprecipitation of Hook3-GFP with KIF1C-Flag, KIF1CΔS-Flag and GFP-Flag from HEK cell lysates. Uncropped gels are provided in Source Data file. **d** Coomassie-stained SDS-PAGE of purified Hook3-SNAPf. The contamination at ~27 kDa is TEV protease used to cleave the protein off the purification tag. **e** Representative kymographs from single-molecule experiments of KIF1C-GFP alone or in the presence of Hook3 labelled with Alexa647. Motility and landing events indicated as in **c**. **f** Quantification of absolute frequency of KIF1C-GFP motors landing on microtubules, n = 116 and 131 MTs, respectively, pooled from three independent experiments. ***p < 0.0005 (t-test). Data are provided in Source Data file

pKIF1C^{RIP2}GFP was replaced with the truncated fragments to create plasmids pKIF1C^{RIP2}ΔCC3-GFP and pKIF1C^{RIP2}ΔS-GFP for mammalian expression. pFastBac-M13-6His-KIF1CΔCC3-GFP and pFastBac-M13-6His-KIF1CΔS-GFP were generated by digesting pKIF1C^{RIP2}ΔCC3-GFP and pKIF1C^{RIP2}ΔS-GFP with BsiWI-BamHI and replacing KIF1C in pFastBac-M13-6his-KIF1CGFP. A human codon-optimised BioID2-HA construct[46] was synthesised to include 5' BamHI and 3' NotI restriction sites allowing direct substitution of the C-terminal GFP of pKIF1C^{RIP2}GFP and pKIF1C^{RIP2}ΔS-GFP with a BioID2-HA tag to generate pKIF1C^{RIP2}-BioID2-HA and pKIF1C^{RIP2}ΔS-BioID2-HA.

The motor-GFP construct (pET22b-KIF1C(1–349)-NTVSVN-GFP-6H) was generated by amplifying the first 349 amino acids of KIF1C using primers UT157 and AS813, digesting with NdeI and NcoI and replacing EB1 in pET22b-mEB1-GFP-6His[47]. The KIF1C stalk construct was cloned into a modified pGEX-6P2 vector (pGEX-6P2-LTLT) which contains an insertion of a tandem TEV cleavage site between the BamHI and EcoRI sites. Amino acids 612 to 922 from human KIF1C were amplified from pKIF1C-2xFlag using AS696 and AS778, digested with EcoRI and NotI, and ligated into pGEX-6P2-LTLT to create pGEX-6P2-LTLT-KIF1C(612–922).

RNAi-protected PTPN21 was made using pHA-PTPN21-WT as a template in a three-step mutagenesis PCR using AS397, AS380 and AS264. The fragment containing the mutation was replaced in pHA-PTPN21-WT using NheI to generate pHA-PTPN21^{RIP}. RNAi-protected inactive PTPN21, pHA-PTPN21^{RIP}C1108S was generated by replacing the fragment containing mutation C1108S in pHA-PTPN21^{RIP} using BsgI.

pFERM expressing HA-tagged PTPN21_{1–377} was generated by digesting pHA-PTPN21^{RIP} and pEGFP-N1 (Clontech) with HindIII and inserting the fragment in pEGFP-N1. As the GFP is not in frame, the FERM domain is expressed from this plasmid with an N-terminal HA-tag only.

To express and purify FERM domains from E. coli, pET22b-HA-PTPN21_{1–381}-6His was generated by amplifying pHA-PTPN21^{RIP} by PCR with primers AS558 and AS592, digesting with NdeI and NotI and replacing EB1 in pET22b-mEB1-6His[47]. For labelling FERM domain, SNAPf-26 was codon optimised for E. coli, synthesised, and ligated into a PCR cloning vector generating pCAPs-SNAPf-26 with the addition of 5' and 3' NdeI and KpnI sites. The HA tag in pET22b-HA-PTPN21_{1–381}-6His was exchanged for SNAPf-26 in a three-fragment ligation where the backbone was digested with PvuI and NdeI or PvuI and KpnI, and the SNAPf-26 insert was digested with NdeI and KpnI.

FERM domain from Ezrin, pET22b-HA-Ezrin_{1–328}-6His was generated by PCR from random-primed cDNA reverse transcribed from RPE1 RNA using primers AS556 and AS557, digesting with NdeI and NotI and replacing in pET22b-mEB1-6His. Hook3 was cloned by PCR from random-primed cDNA reverse transcribed from RPE1 RNA using primers AS690 and AS691, which was incorporated between the EcoRI and BamHI site of pKIF1C^{RIP2}GFP, replacing the KIF1C and generating pKan-CMV-Hook3-GFP. A multi-tag version of pFastBacM13 (Invitrogen) was created by ligating an insect-cell codon optimised synthesised cDNA for 8His-ZZ-LTLT-BICDR1-SNAPf between the RsrII and NotI sites creating pFastBacM13-8His-ZZ-LTLT-BICDR1-SNAPf. Hook3 was cloned into this by PCR from random-primed cDNA reverse transcribed from RPE1 RNA using

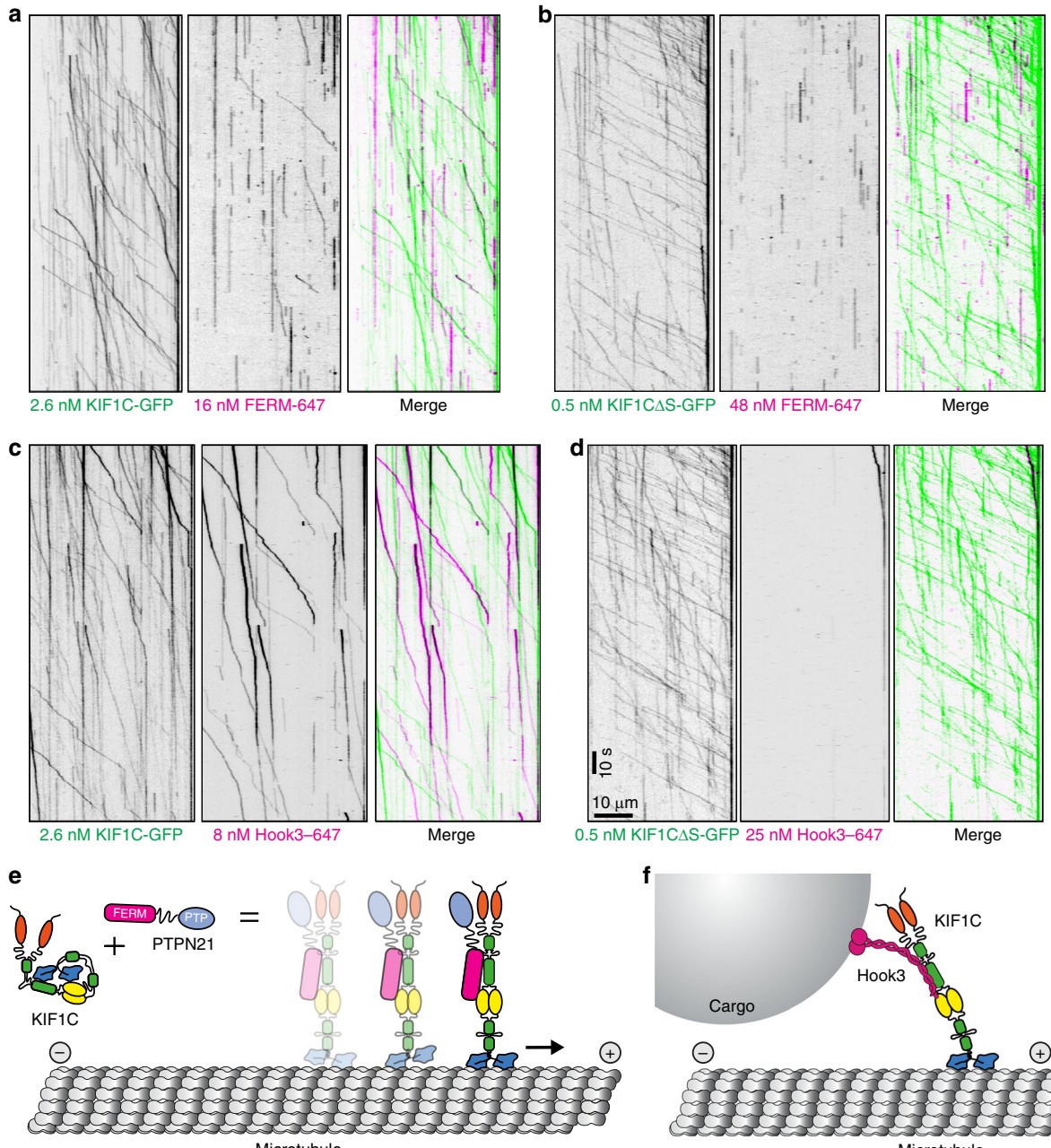

**Fig. 8** Activation and cotransport with KIF1C. **a–d** Both PTPN21-FERM domain and Hook3 cargo adapter are co-transported with KIF1C if the stalk domain is present. **e, f** Model for KIF1C activation. KIF1C is an autoinhibited dimer in solution. Intramolecular interactions between stalk and motor domain prevent the inhibited motor from interacting with microtubules. Upon binding of PTPN21, the intramolecular interactions are released, KIF1C can engage with microtubules and will step processively towards the plus end. The cargo adapter Hook3 also binds to the stalk of KIF1C and activates it, thereby mediating cargo-activated transport along microtubules

primers AS689 and AS691 which allowed direct ligation into the multi-tag *pFastBacM13* vector creating *pFastBacM13-8His-ZZ-LTLT-Hook3-SNAPf*, which was subsequently used for production of a recombinant baculovirus.

Duplex siRNA oligos targeting KIF1C, KIF16B and PTPN21 were custom synthesised by Sigma and were: siControl 5′-GGACCUGGAGGUCUGCUGU-[dT]-[dT]-3′ (ref. [3]), siKIF1C-2 (targeting both rat and human KIF1C) 5′-GUGA GCUAUAUGGAGAUCU-[dA]-[dT]-3′ (refs. [3,5]), siPTPN21 (targeting both rat and human PTPN21) 5′-UUCAGCCUCUGGUACUACA-[dT]-[dT]-3′, siKIF16B (targeting rat KIF16B) 5′-GAACUACAGCGACGUGGAG-[dT]-[dT]-3′.

**Recombinant protein purification**. For baculovirus expression, pFastBac-M13-6His-KIF1CGFP, pFastBac-M13-6His-KIF1CΔCC3-GFP, pFastBac-M13-6His-KIF1CΔS-GFP and pFastBacM13-8His-ZZ-LTLT-Hook3-SNAPf plasmids were transformed into DH10BacYFP competent cells[48] and plated on LB-Agar

supplemented with 30 μg/ml kanamycin (#K4000, Sigma), 7 μg/ml gentamycin (#G1372, Sigma), 10 μg/ml tetracycline (#T3258, Sigma, 40 μg/ml Isopropyl β-D-1-thiogalactopyranoside (IPTG, #MB1008, Melford) and 100 μg/ml X-Gal (#MB1001, Melford). Positive transformants (white colonies) were screened by PCR using M13 forward and reverse primers for the integration into the viral genome. The bacmid DNA was isolated from the positive transformants by the alkaline lysis method and transfected into SF9 cells (Invitrogen) with Escort IV (#L-3287, Sigma) according to the manufacturer's protocols. After 5–7 days, the virus (passage 1, P1) is harvested by centrifugation at $300 \times g$ for 5 min in a swing out 5804 S-4-72 rotor (Eppendorf). Baculovirus infected insect cell (BIIC) stocks were made by infecting SF9 cells with P1 virus and freezing cells before lysis (typically around 36 h) in a 1° cooling rate rack (#NU200 Nalgene) at −80 °C. P1 virus was propagated to passage 2 (P2) by infecting 50 ml of SF9 (VWR, #EM71104-3) culture and harvesting after 5–7 days as described above[49]. For large-scale expression, 500 ml of SF9 cells at a density of $1–1.5 \times 10^6$ cells/ml were infected with one vial of BIIC or P2 virus. Cells

were harvested when 90% infection rate was achieved as observed by YFP fluorescence, typically between 48 and 72 h. Cells were pelleted at $252 \times g$ in a SLA-3000 rotor (Thermo Scientific) for 20 min. For purification of recombinant KIF1C-GFP, KIF1CΔCC3-GFP and KIF1CΔS-GFP, the pellet was resuspended in 4 ml of SF9 lysis buffer (50 mM Sodium phosphate pH 7.5, 150 mM NaCl, 20 mM Imidazole, 0.1% Tween 20, 1.5 mM $MgCl_2$) per gram of cell pellet, supplemented with 0.1 mM ATP and cOmplete protease inhibitor cocktail (#05056489001, Roche) and lysed using a douncer (#885301, Kontes) with 20 strokes. Lysates were then cleared by centrifugation at $38,000 \times g$ in a SS-34 rotor (Sorvall) for 30 min, or $200,000 \times g$ for 40 min in a T865 rotor (Sorvall). SP Sepharose beads (#17-0729-01, GE Healthcare) were equilibrated with the lysis buffer and the cleared lysate obtained is mixed with the beads and batch bound for 1 h. Next, the beads were loaded onto a 5 ml disposable polypropylene gravity column (#29922, Thermo Scientific) and washed with at least 10 CV SP wash buffer (50 mM sodium phosphate pH 7.5, 150 mM NaCl) and eluted with SP elution buffer (50 mM sodium phosphate pH 7.5, 300 mM NaCl). The peak fractions obtained were pooled and diluted with Ni-NTA lysis buffer (50 mM sodium phosphate pH 7.5, 150 mM NaCl, 20 mM Imidazole, 10% glycerol) and batch bound to Ni-NTA beads (#30230, Qiagen) for 1 h. The beads were loaded onto a gravity column and washed with at least 10 CV of Ni-NTA wash buffer (50 mM sodium phosphate pH 7.5, 150 mM NaCl, 50 mM Imidazole and 10% glycerol) and eluted with Ni-NTA elution buffer (50 mM sodium phosphate pH 7.5, 150 mM NaCl, 150 mM Imidazole, 0.1 mM ATP and 10% glycerol). The peak fractions were run on a SDS-PAGE gel for visualisation and protein was aliquoted, flash frozen and stored in liquid nitrogen.

Hook3-647 was purified and labelled in a two-step process utilising both the His and ZZ affinity tags at 4 °C. A pellet corresponding to 500 ml insect culture was resuspended in 40 ml Adaptor Lysis Buffer (50 mM HEPES pH 7.2, 150 mM NaCl, 20 mM imidazole, 1× cOmplete protease inhibitor cocktail, 1 mM DTT) and lysed with 20 strokes in a douncer. Lysates were cleared of insoluble material by centrifugation at $50,000 \times g$ for 40 min in a Sorvall T-865 rotor. The cleared lysate was bound to 2 ml Ni-NTA beads for 2 h, and subsequently washed with 200 CV of lysis buffer followed by 200 CV lysis buffer supplemented with 60 mM imidazole. Protein was eluted in 5 CV of lysis buffer containing 350 mM Imidazole, and the eluate was batch bound to 1 ml IgG Sepharose (#17-0969-01, GE Healthcare Life Sciences) for 2 h. Beads were briefly washed with TEV cleavage buffer (50 mM Tris-HCl pH 7.4, 148 mM KAc, 2 mM MgAc, 1 mM EGTA, 10% (v/v) glycerol, 1 mM DTT) before being collected in a 1.5 ml Eppendorf tube and incubated with 3.5 µM SNAP-Surface Alexa Fluor 647 substrate (#S9136S, New England Biolabs) for 2 h. Beads were washed with 200 CV TEV cleavage buffer and resuspended in a 1.5 ml Eppendorf tube containing 50 µg/ml TEV protease, which was allowed to digest the protein off of beads overnight at 4 °C. Finally, eluate was collected from the beads, concentrated to between 5 and 10 µM in a Centrisart I 20 kDa ultrafiltration device (#13249-E, Sartorius), flash frozen and stored in liquid nitrogen. Labelling ratio was 1.05 for Hook3-647.

Bacterial expression plasmids, pET22b-HA-PTPN21$_{1-381}$-6His and pET22b-HA-Ezrin$_{1-328}$-6His were transformed into BL21 DE3 (Invitrogen), pET22b-KIF1C (1–349)-NTVSVN-GFP-6H, pET22b-SNAPf26-PTPN21$_{1-381}$-6His, pGEX-6P2 and pGEX-6P2-LTLT-KIF1C(612–922) were transformed into BL21 CodonPlus RIL (Agilent). Single colonies were grown overnight as 3 ml starter cultures. The starter cultures were diluted 1:100 and grown in 200 ml of 2× YT (yeast extract, tryptone 16 g) at 37 °C, 180 r.p.m. until they reached an OD$_{600nm}$ of 0.5. Expression was induced with 0.5 mM or 1 mM IPTG and incubated at 16 °C or 18 °C overnight.

For purification of recombinant PTPN21-FERM, SNAPf26-PTPN21-FERM and Ezrin-FERM, cells were harvested at $1500 \times g$ in an SLA-3000 rotor and resuspended in bacterial lysis buffer (50 mM sodium phosphate pH 7.5, 50 mM NaCl, 20 mM Imidazole, 1.5 mM $MgCl_2$) supplemented with 1 mM phenylmethanesulfonyl fluoride (PMSF) (#MB2001; Melford), 1 mg/ml lysozyme and sonicated at 50% amplitude for 30 s in 10 s on off cycle repeated thrice. Extracts were cleared and purified using SP and Ni-NTA bead as described for KIF1C-GFP above. Labelling of SNAPf26-PTPN21-FERM was performed on the Ni-NTA beads by adding 3.5 µM SNAP-Surface Alexa Fluor 647 substrate and incubating for 1 h at 4 °C. Beads were washed with bacterial Ni-NTA wash buffer (50 mM sodium phosphate pH 7.5, 50 mM NaCl, 50 mM Imidazole and 10% glycerol) to remove unbound dye and eluted with bacterial Ni-NTA elution buffer (50 mM sodium phosphate pH 7.5, 50 mM NaCl, 150 mM Imidazole and 10% glycerol). Labelling ratio was 0.4 for FERM-647. Purified proteins were flash frozen and stored in liquid nitrogen. For purification of motor-GFP (i.e. KIF1C(1–349)-GFP-6H), cells were lysed in Motor Lysis Buffer (50 mM Tris pH 7.4, 300 mM NaCl, 20 mM Imidazole, 1 mM DTT, 1 mM $MgCl_2$, 0.1 mM ATP, 0.05% Tween 20) supplemented with 2.5 mg/ml lysozyme and cleared of insoluble material by centrifugation at $100,000 \times g$ for 30 min at 4 °C in a Sorvall T865 rotor. Protein was batch bound to 1.5 ml Ni-NTA agarose for 1.5 h at 4 °C before being washed with 450 ml lysis buffer followed by 200 ml washing buffer (lysis buffer with 40 mM Imidazole). 0.5 CV elutions were collected using lysis buffer supplemented with 500 mM imidazole and 10% glycerol. To purify GST-stalk (i.e. GST-KIF1C (612–922)) and GST control, bacterial cells were lysed in Tail Lysis Buffer (50 mM Tris pH 7.4, 300 mM NaCl, 0.05% Tween 20, 1 mM DTT) supplemented with 2.5 mg/ml lysozyme, 1× cOmplete protease inhibitor and 2 mM PMSF. Lysate was cleared by centrifugation at $100,000 \times g$ for 30 min at 4 °C in a Sorvall T865 rotor. Cleared lysate was bound to 1.5 ml SuperGlu GSH resin (#SuperGluA, Generon) for 1.5 h at 4 °C, loaded into a column and washed with 450 ml lysis buffer,

followed by 250 ml Tail Wash Buffer (50 mM Tris pH 7.4, 500 mM NaCl, 0.05% Tween 20, 1 mM DTT). Elutions of 0.5 CV were collected using lysis buffer supplemented with 20 mM reduced glutathione and 10% glycerol. Purified proteins were flash frozen and stored in liquid nitrogen.

**Hydrodynamic analysis.** Size exclusion chromatography was carried out using Superdex 16/60 200 pg (#28989335, GE Healthcare) column on an AKTApurifier10 FPLC controlled by UNICORN 5.11 (GE Healthcare). The column was equilibrated with the SEC Buffer (35 mM sodium phosphate pH 7.5, 150 mM NaCl, 1.5 mM $MgCl_2$). Two hundred microlitres of ~0.5 mg/ml KIF1C-GFP or KIF1C-Flag was injected into the column and 0.5 ml fractions were collected using the fraction collector. GFP fluorescence in the fractions was measured using Nanodrop 3300 Fluorospectrometer (Thermo Scientific) to determine peak positions. One hundred microlitres of 5 mg/ml standard proteins were injected individually. The Stokes radius for the standard proteins used were as follows thyroglobulin (#T9145, Sigma) 8.5 nm[50], apoferritin (#A3660, Sigma) 6.1 nm[51], catalase (#C9322, Sigma) 5.2 nm[52], bovine serum albumin (#A7906, Sigma) 3.48 nm[53]. Log $R_s$ vs. (elution volume-void volume) for standard proteins was plotted and stokes radius $R_s$ for KIF1C was determined from the slope of the line, $y = (-0.014 \pm 0.002) \times x + (0.97 \pm 0.04)$. Three independent repeats for KIF1C-GFP were carried out at 500 mM NaCl and two at 150 mM NaCl. KIF1C-Flag was carried out once at 150 mM NaCl.

Five millimetres of 10–40% vol/vol glycerol gradients were made using the Gradient master (#108, Biocomp) in 35 mM sodium phosphate pH 7.5, 1.5 mM $MgCl_2$, 0.1 mM ATP, 1 mM EGTA at different NaCl concentration (50, 150, 250, 350, 500 mM NaCl). One hundred microlitres of ~0.25 mg/ml KIF1C-GFP or KIF1C-Flag was loaded on top of the gradient and the samples were centrifuged at $364,496 \times g$ in SW55Ti (Beckman Coulter) for 16 h at 4 °C. Gradients were fractionated by carefully pipetting 200 µl aliquots from the top. Absorbance at 280 nm was measured in Nanodrop 2000c Spectrophotometer and fluorescence using a Nanodrop 3300 Fluorospectrometer (Thermo Scientific) to determine peak positions. One hundred microlitres of 5 mg/ml standard proteins were loaded individually on separate gradients. The samples were processed for SDS-PAGE and immunoblotting with anti-KIF1C primary (1:4000, #AKIN11, Lot 011, Cytoskeleton) and anti-rabbit IgG HRP conjugate secondary (1:4000, #W401B, Lot 237671, Promega) antibodies. The sedimentation coefficient of standard proteins were: apoferritin 17.6S[54], catalase 11.3S[52], glucose oxidase (#G7141, Sigma) 8S[55], bovine serum albumin 4.6S[53]. Sedimentation coefficient vs. sedimentation volume for standard proteins was plotted and sedimentation coefficient for KIF1C at different salt concentration was calculated from the slope of the line, $y = (5.66 \pm 0.49)^*x^* - (3.92 \pm 1.31)$. Independent repeats for KIF1C-GFP were carried out thrice (500 mM NaCl and 50 mM NaCl), twice (150 and 250 mM) and once (350 mM), respectively. KIF1C-Flag was carried out twice at 150 mM NaCl.

The molecular weight was calculated from stokes radius and sedimentation coefficient using equation $M_r = 4205 \times R_S \times S$ as described in ref. [56]. The frictional coefficient was calculated from the following formula, $f/f_{min} = S_{max}/S$ with $S_{max} = 0.00361 \cdot M^{2/3}$ (ref. [56]) using M = 308,000 Da for KIF1C-GFP dimer and $S$ is the sedimentation coefficient determined from glycerol gradient centrifugation.

**Crosslinking mass spectrometry.** Two cross-linkers, BS3 (bis (sulpho-succinimidyl) suberate) (#21580, Thermo Scientific) and EDC (1-ethyl-3-(3-dimethylaminopropyl) carbodiimide) (#22980, Thermo Scientific) were used to analyse protein interactions using crosslink mass spectrometry. Five millimolar of cross-linker was freshly prepared in MilliQ water and was mixed in the ratio of 1:2 with the protein(s) of interest by pipetting. KIF1C-GFP concentration was 1 µM when crosslinked alone. For crosslinking in the presence of PTPN21-FERM, both proteins were at 0.5 µM. Final concentration NaCl in the crosslinking reaction was 150 mM or increased to 500 mM for high salt sample. For EDC crosslinking, 3 mM $N$-hydroxysulfosuccinimide (#24510, Thermo Scientific) was added to the reaction with EDC to improve efficiency of the crosslinking. The reaction was incubated shaking at 400 r.p.m. for an hour at room temperature and then quenched with 50 mM Tris-HCl pH 7.5. Next, the protein was diluted in equal volume of 50 mM ammonium bicarbonate (#A6141, Sigma) and reduced with 1 mM DTT (#MB1015, Melford) for 60 min at room temperature. The sample was then alkylated with 5.5 mM iodoacetamide (#I6709, Sigma) for 20 min in the dark at room temperature and digested using 1 µg trypsin (sequencing grade; #V5111, Promega) per 100 µg of protein overnight at 37 °C. The crosslinked peptides were desalted using C18 stage tips. Twenty microlitre samples were then analysed by nano LC-ESI-MS/MS using UltiMate® 3000 HPLC series using Nano Series™ Standard Columns for separation. A linear gradient from 4% to 35% solvent B (0.1% formic acid in acetonitrile) was applied over 30 min, followed by a step change 35–70% solvent B for 20 min and 70 to 90% solvent B for 30 min. Peptides were directly eluted (~300 nl/min) via a Triversa Nanomate nanospray source into a Orbitrap Fusion mass spectrometer (Thermo Scientific). Positive ion survey scans of peptide precursors from 375 to 1500$m/z$ were performed at 120 K resolution (at 200$m/z$) with automatic gain control $4 \times 10^5$. Precursor ions with charge state 2–7 were isolated and subjected to HCD fragmentation in the Orbitrap at 30 K resolution or the Ion trap at 120 K. MS/MS analysis was performed using collision energy of 33%, automatic gain control $1 \times 10^4$ and max injection time of 200 ms. The dynamic exclusion duration was set to 40 s with a 10 ppm tolerance for the

selected precursor and its isotopes. Monoisotopic precursor selection was turned on. The instrument was run in top speed mode with 2 s cycles.

Raw data files were converted to .mgf format using the ProteoWizard msconvert toolkit[57]. Sequences are visualised using Scaffold (Proteome software) for percentage coverage and purity followed by analysis using StavroX[22]. Crosslinked peptides were identified using the StavroX software with appropriately defined parameters for the crosslinker used. A crosslinked peptide was considered as valid if it passed the 5% false discovery rate (FDR) cut-off along with a StavroX score of at least 100, which is based on: (i) the presence of ion series fragmentation for both peptides, specifically those fragment ions that include the crosslinker and the attached second peptide, (ii) the proximity of observed fragmentation ion mass to expected fragment ion mass, (iii) number of ion fragments for b or y ions in the a peptide, (iv) number of unidentified high intensity signals in the spectra. The MS/MS spectra were also manually inspected and only those crosslinks were accepted for which fragmentation ions were observed for both peptides and three or more fragments for b or y ions in the alpha peptide were required to match. Any crosslinks of continuous peptides, which could indicate intradimer interactions, were rejected as these could not be distinguished from partially cleaved peptides that have been modified by the crosslinker. Once a significant crosslink was identified in at least one sample, we used the information on retention time and mass of the precursor ion to verify the presence of the crosslinked peptide in other samples.

Label-free quantification of crosslinked precursor ions and unmodified peptides was done using MaxQuant (V1.5.5.1). The LFQ module calculates the integrated peak area of the precursor ion based on a retention time window of 4 min and ppm error window of 10 ppm to quantify the presence and absence of a precursor ion of interest in different mass spectrometry output files. LFQ intensity data for each sample were normalised to the sum of unmodified and modified peptides containing LKEGANINK. BS3 experiments were repeated 3–5 times and EDC experiments 2–3 times for each sample.

**Microscale thermophoresis**. GST-stalk was rebuffered into MST buffer (50 mM Tris pH 7.4, 150 mM NaCl, 10 mM MgCl2, 0.05 % Tween- 20) using Zeba Spin Desalting Columns (Thermo Scientific). The resulting 37 μM stock solution was used to prepare a twofold dilution series. Motor-GFP was diluted in MST buffer to a final concentration of 250 nM, 2 μl of which was mixed with 8 μl of each GST-stalk dilution. Capillaries were filled with 10 μl protein mix and thermophoresis performed in a Monolith NT.115 (Nanotemper) using 30% Nano Blue detection and 40% MST power at 25 °C. The experiment was repeated three times. Traces were analysed in 0.5 and 1.5 s heating window and the Kd model was fitted using MO.Affinity Analysis software (Nanotemper).

**BioID protein interaction analysis**. For each BioID experiment, three 14.5 cm dishes of RPE1 cells were grown, and each was transfected at 90% confluency with 20 μg DNA diluted in 1 ml Optimem (#31985062, Fisher) with 60 μg poly-ethylenimine (PEI, #408727, Sigma). After 24 h, media was replaced and supplemented with 50 μM biotin (#8.51209, Sigma). After a further 12 h, cells were harvested by trypsinisation, washed with PBS and lysed in RIPA buffer (#9806S, New England Biolabs) supplemented with 250 U of benzonase (#E1014, Sigma). Lysate was cleared by centrifugation in a tabletop centrifuge at 20,238 ×g for 30 min at 4 °C (#5417R, Eppendorf), and bound to 50 μl streptavidin-coated Dynabeads (#65601, Invitrogen) for 1.5 h, washing away unbound proteins with 3 × 1 ml of PBS. Beads were resuspended in 45 μl 50 mM ammonium bicarbonate, reduced by adjusting to a final concentration of 10 mM TCEP, and subsequently alkylated by adjusting to 40 mM chloroacetamide (#C0267, Sigma), incubating at 70 °C for 5 min. Proteins were digested off of the beads with 0.5 μg Trypsin (#V5111, Promega) at 37 °C overnight. Peptides were separated from beads by filtration through a 0.22 μm tube filter (#CLS8169, Sigma) and desalted using C18 stage tips. Twenty microlitres of peptides were then analysed by nano LC-ESI-MS/MS in an Ultimate 3000/Orbitrap Fusion. Hits were automatically identified, thresholded and viewed in Scaffolds software. We performed three independent experiments of which one had three technical replicated. Statistical analysis was performed in Scaffolds using a Fisher's exact test with Benjamini–Hochberg correction, which resulted in a significance threshold of 0.0019 for a 5% FDR.

**Cell culture**. A7r5 rat vascular smooth muscle cells (ATCC, #CRL-1444) were grown and maintained in low-glucose (1000 mg/l) Dulbecco's modified Eagle's medium (DMEM), supplemented with 10% foetal bovine serum at 37 °C and 5% CO2. For siRNA oligonucleotide transfection, HiPerFect (#301704 Qiagen) was used according to the manufacturer's protocol. Rescue DNA plasmids were transfected using Fugene6 (#E2691, Promega) 24 h after siRNA transfection, cells seeded onto 16 mm glass coverslips (#1232–3148 Fisherbrand) coated for 24 h with 10 μg/ml fibronectin (#F1141 Sigma) and analysed 72 h1 after siRNA transfection. Podosome formation was stimulated in A7r5 cells by treatment with 5 μM PDBu (Phorbol 12,13-dibutyrate; #P1269 Sigma) for 1 h. For inhibition of Myosin IIA contractile activity, Y27632 Rock inhibitor (#Y0503 Sigma) or Blebbistatin (#B0560 Sigma) were added at various concentrations as indicated in figure labels at the same time as PDBu. DMSO (#D2438, Sigma) was used as a negative control.

The RPE1 α5-integrin-GFP stable cell line was established as follows: hTERT RPE1 cells (Clontech) were transfected with pα5-integrin-GFP[44] followed by selection with 500 μg/ml Geneticin (#G8168 Sigma) and expansion of single colonies that were screened using fluorescence microscopy. The α5-integrin-GFP cell line was grown in RPE1 growth medium (DMEM/Nutrient F-12 Ham (#D6421 Sigma), 10% FBS (Sigma), 2 mM L-glutamine (Sigma), 100 U/ml penicillin (Sigma), 100 μg/ml streptomycin (Sigma) and 2.3 g/l sodium bicarbonate (#S8761 Sigma)) supplemented with 500 μg/ml Geneticin (Sigma). RPE1 cells were transfected with siRNA using Oligofectamine (#12252011 Invitrogen), with DNA using Fugene6 and analysed 72 h after siRNA transfection.

Hippocampi were dissected from P1 or P2 mice brains and incubated with papain (#P4762 Sigma) in cold dissection medium (Earl's buffered salt solution EBSS #14155 Gibco), 10 mM HEPES (#H0887 Sigma) and 100 U/ml pen-strep) for 15 min at 37 °C. After digestion, the hippocampi were spun at 300 × g for 2 min in a swing out 5804 S-4-72 rotor, washed (10 ml) and resuspended (1 ml) in plating media (modified eagle medium (MEM) (#51200–046 Gibco) supplemented with 100 U/ml pen-strep, 1% N2 (#17502-048, Gibco), 10% horse serum (#26050-039, Gibco), 20 mM glucose, 1 mM sodium pyruvate (#S8636, Sigma) and 25 mM HEPES). Cells were plated on laminin (#L2020, Sigma)/poly-D-lysine (#P0899, Sigma)-coated coverslips and fed by replacing 50% of the medium with plating medium according to Granseth et al.[58]. Neurons were transfected at DIV4 or DIV5 with lipofectamine 2000 (#11668027, Thermo Scientific) according to the manufacturer's protocol and imaged the next day.

**Fixed cell imaging**. Cells were fixed for 15 min with 4% paraformaldehyde (#15714, Electron Microscopy Sciences) diluted in Cytoskeleton Buffer (10 mM MES pH 6.1, 138 mM KCl, 3 mM MgCl2, 2 mM EGTA, 0.32 M sucrose). Fixed cells were incubated for 2 min with 0.1% Triton X-100 (#T8787, Fisher Scientific) diluted in PBS. Coverslips were washed with PBS and incubated for 30 min with 0.5% BSA diluted in PBS/0.1%Tween (PBST). Cells were stained for an hour at room temperature or overnight at 4 °C with 1:100 anti-cortactin 4F11, (#05-180, Lot 2290226, Millipore) primary antibodies diluted in 0.5% BSA-PBST solution. Coverslips were washed with PBST and incubated for 45 min at room temperature with 1:300 anti-mouse IgG 647 conjugate (#A31571, Lot 47735A, Molecular Probes) secondary antibodies and 1:1000 Acti-stain 555 phalloidin (#PHDH1-A, Cytoskeleton) diluted in 0.5% BSA in PBST. Nuclei were stained with 5 μg/ml DAPI (#D9542, Sigma) for 1 min and coverslips washed with PBST prior to mounting on glass slides with Vectashield (#H1000, Vector Laboratories).

Cells were imaged using a Deltavision Elite Wide-field microscope and Z-stacks of individual cells were acquired using a ×40 NA 1.49 objective and a Z-spacing of 0.2 μm.

To determine the number of podosomes formed in each cell, images of the cortactin channel were first transformed in a Z-projection. Z-projection images were then segmented using the ImageProAnalyzer 7 software by applying a threshold and a minimal size filter of 16 pixels (2.58 μm²). Individual objects identified in the cortactin Z-projection were then visually compared to the actin channel to confirm they are podosomes, removed if the cortactin staining did not coincide with an actin spot and podosome clusters were split into individual podosomes. All experiments were repeated three times with 30 cells being analysed for each condition in each experiment.

**Live cell imaging**. Live cells were imaged using a ×60 oil NA 1.4 objective on an Olympus Deltavision microscope (Applied Precision, LLC) equipped with eGFP, mCherry filter sets and a CoolSNAP HQ2 camera (Roper Scientific) under the control of SoftWorx (Applied Precision). The environment was maintained at 37 °C and 5% CO2 using a stage-top incubator (Tokai Hit) and a weather station (Precision control).

RPE1 α5-integrin-GFP cells were seeded 24 h before imaging into quadrant glass-bottom dishes (#627975, Greiner) coated with 10 μg/ml Fibronectin. Cells with tails were selected and the mid region of the tail was bleached using the 488 nm laser. Vesicle movement was imaged for 200 time points at 0.7–1.0 s per frame. α5-integrin-GFP trafficking was analysed using ImageJ. Kymographs were generated from parallel 21 pixel wide lines covering the area of the tail. Vesicle movement was extracted manually from all kymographs using the segmented line tool. Vesicles moving less than a total of 1.5 μm over the period of imaging were classified as stationary.

Primary mouse hippocampal neurons transfected with pNPY-RFP and either pFlag (as control) or pFERM and imaged at DIV5 or DIV6. Images were acquired with 500 ms exposure every 1.5 s for 160 s. The frequency of NPY-positive vesicles passing through a location in the neurite in anterograde and retrograde direction was determined manually from kymographs.

RPE1 cells co-transfected with pKIF1C-mCherry and either pKIF1C-GFP, pKIF1CΔCC3-GFP or pKIF1CΔS-GFP were imaged 36 h post transfection. Images were acquired with 500 ms exposure in the eGFP channel and 1 s exposure in the mCherry channel. To determine the ratio of enrichment at the tail, a region of interest was drawn manually surrounding the accumulation observed and the mean intensity was measured at the tail, in the cytoplasm near the nucleus and the image background outside the cell in both channels. The ratio of KIF1C enrichment at the tail was calculated as $I_{tail} - I_{background}/I_{cytoplasm} - I_{background}$ for GFP vs. mCherry channel.

**Co-immunoprecipitation.** HEK293 cells (Agilent, #240073) were seeded at a density of $1.5 \times 10^6$ cells onto one 10 cm dish 24 h before transfection with *pKIF1C^RIP1-2Flag*, *pKIF1CΔS-GFP* or *pGFP-4Flag* and either *pHook3-GFP* or *pHA-PTPN21^RIP*. Two micrograms of DNA was mixed with 6 μl 1 mg/ml PEI (Sigma) in 200 μl PBS, incubated for 15 min and then added onto cells for transfection. Transiently transfected cells were collected 20–24 h after transfection by decanting the media and washing the cells off the dish with 5 ml ice cold 1× PBS. Cells were collected by centrifugation at $300 \times g$ for 3 min and washed twice with 1× PBS. The cell pellets were lysed in 130 μl of dynein lysis buffer (DLB: 30 mM HEPES, pH 7.4; 50 mM KOAc; 2 mM MgOAc; 1 mM EGTA, pH 7.5; 10% glycerol) supplemented with 1 mM DTT, 0.2% Triton X-100, 1× protease inhibitor cocktail (Expedion) by sonication in the ice- water bath of the Bioruptor (Diagenode) at medium setting for 5 min (30 s on, 30 s off cycle). Lysates were cleared at maximum speed in Eppendorf microcentrifuge (5417 R) at 4 °C for 15 min. For each reaction, 10 μl of Dynabeads Protein G (Invitrogen 100.03D) were incubated with 1 μg anti-Flag antibodies (#F3165, Lot 058K6113, Sigma) for 20 min at room temperature and then washed three times with 100 μl DLB. One hundred and twenty microlitres of cell lysate was added to the beads and incubated for 25 min at 4 °C. The beads were washed seven times with 1 ml DLB. To recover bound proteins, beads were resuspended in 35 μl 1× Laemmli buffer and incubated at 95 °C for 2 min. Samples were separated over 10% polyacrylamide gels followed by immunoblotting. Western blots were detected using 1:4000 anti-KIF1C (#AKIN11, Lot 011, Cytoskeleton Inc.), 1:5000 anti-HA (C29F4) (#3724 S, Lot 9, Cell Signalling Technologies) and 1:3000 anti-Hook3 (#15457-1-AP, Proteintech) as primary antibodies and anti-rabbit-HRP (#W401B, Lot 237671, Promega). Detection with SuperSignal West Pico Plus substrate (Pierce) was followed by imaging chemiluminescence in G-Box (Syngene) or using blue-sensitive film.

**Single-molecule motility and binding assays.** Microtubules were assembled from 8 μl of 3.4 mg/ml unlabelled pig tubulin, 0.2 μl of 1 mg/ml HiLyte Fluor 670 tubulin (#TL670M, Cytoskeleton) and 0.5 μl of 0.5 mg/ml biotin tubulin (#T333P, Cytoskeleton) in MRB80 (80 mM PIPES pH 6.8, 4 mM MgCl₂, 1 mM EGTA, 1 mM DTT). The mixture was incubated on ice for 5 min before adding 8.5 μl of polymerisation buffer (2× BRB80 buffer plus 20% (v/v) DMSO and 2 mM MgGTP). Microtubules were polymerised at 37 °C for 30–60 min. The sample was diluted with 100 μl MT-buffer (MRB80 plus 30 μM paclitaxel). Unincorporated tubulin was removed by pelleting microtubules at $20,238 \times g$ for 8.5 min at room temperature, washing the pellet with 100 μl MT-buffer and re-pelleting as before. The microtubule pellet was resuspended in 50 μl of MT-buffer and stored at RT covered from light for at least half a day (maximum 3 days) before use.

Coverslips (22 × 22) were cleaned by incubating in 2.3 M hydrochloric acid overnight at 60 °C. The next day, coverslips were washed with Millipore water and sonicated at 60 °C for 5 min. The wash cycle was repeated five times. The coverslips were dried using a Spin Clean (Technical video) and plasma cleaned using Henniker plasma clean (HPT-200) for 3 min. Flow chambers were made using clean glass slides (Menzel Gläser Superfrost Plus, Thermo Scientific) and double-sided sticky tape (Scotch 3 M) by placing the cleaned coverslip on the sticky tape creating a 100 μm deep flow chamber. The surface was coated with (0.2 mg/ml) PLL(20)-g[3.5]-PEG(2)/PEG(3.4)-Biotin (50%) (#PLL(20)-g[3.5]-PEG(2)/PEGbi, Susos AG). Biotin-647-microtubules were attached to this surface with streptavidin (0.625 mg/ml) (#S4762 Sigma) and the surface was blocked with κ-casein (1 mg/ml) (#C0406 Sigma).

KIF1C-GFP, PTPN21-FERM and Ezrin-FERM were pre-spun at $100,000 \times g$ for 10 min in an Airfuge (Beckman Coulter). Six hundred picomolar KIF1C-GFP was incubated with either Ni-NTA elution buffer or 125 nM PTPN21-FERM or 125 nM Ezrin-FERM for 15 min at room temperature. The complex was then diluted in the motility mix (MRB80 supplemented with 1 mM ATP, 5 mM phosphocreatine (#P7936, Sigma), 7 U/ml creatine phosphokinase (#C3755 Sigma), 0.2 mg/ml catalase, 0.4 mg/ml glucose oxidase, 4 mM DTT, 50 mM glucose (#G8270, Sigma), 25 mM KCl, 100 μM taxol, 0.2 mg/ml κ-casein) and flown into the chamber. Experiments with KIF1CΔCC3-GFP and KIF1CΔS-GFP were performed in the same way. To verify that comparable amount of motors were added to the assay, the fluorescence intensity of KIF1C-GFP, KIF1CΔCC3-GFP and KIF1CΔS-GFP solutions were measured prior to incubation and flowing into the chamber. KIF1C and Hook3 experiments were prepared using 8 nM Kif1C-GFP and 25 nM Hook3-647 in the same motility mix, but replaced MRB80 with TIRF assay buffer (30 mM HEPES-KOH pH 7.2, 5 mM MgSO₄, 1 mM EGTA, 1 mM DTT) as this has previously been described for Hook3-based dynein motility[28].

Chambers were observed on an Olympus TIRF system using a ×100 NA 1.49 objective, 488 and 640 nm laser lines, an ImageEM emCCD camera (Hamamatsu Photonics) under the control of xCellence software (Olympus), an environmental chamber maintained at 25 °C (Okolab, Ottaviano, Italy).

For single-molecule motility assays with KIF1C-GFP, KIF1CΔCC3-GFP, KIF1CΔS-GFP and FERM domains images were acquired at 30% laser power every 100 ms for 180 s at an exposure of 60 ms for 488 nm laser line, whereas for experiments with KIF1C-GFP and Hook3, images were acquired at 280 ms for 250 s at an exposure of 150 ms for 488 nm (30% laser power) and 60 ms for 647 nm (10% laser power) laser lines.

For cotransport assay, KIF1C-GFP or KIF1CΔS-GFP was incubated with either Hook3-SNAPf647 or 647SNAPf-FERM at 20× the final concentration before being

diluted into imaging buffer. Images were taken once every 275 ms using 50 ms exposure with a 647 nm laser line followed by 150 ms exposure with a 488 nm laser line, using laser powers of 18% and 21%, respectively.

Images were analysed by generating kymographs from each microtubule using the ImageJ plugin by Arne Seitz (https://biop.epfl.ch/TOOL_KYMOGRAPH.html) and manually identifying runs by scoring the kymographs for moving and static motors. The speed and run length was calculated from the length and slope of each run using a custom ImageJ macro. Blind analysis was carried out to determine landing rates and frequency of running motors in all data sets.

For TIRF-based microtubule binding assays, 80 nM motor-GFP was mixed with either 20 μM GST-stalk or the same amount of GST as a control. These mixtures were diluted 1:20 into motility mix as described for KIF1C-Hook3 single-molecule experiments, yielding a final protein concentration of 4 nM motor domain to 1 μM tail or GST control. The motility mixture was flown into chambers prepared with unlabelled GDP-taxol microtubules, and microtubule intensities in ten random fields-of-view were captured with an exposure time of 400 ms and 21.4% 488 laser power. Control and experimental chambers are prepared in parallel, and we alternated which chamber was imaged first to account for time effects. For analysis, microtubules were traced by hand with a line ROI in ImageJ, and the mean intensity was taken. A region close to the microtubule was selected as the local background, and subtracted from the mean to generate a local-background corrected intensity.

**Bleach step analysis.** A flow chamber was prepared as described above with biotin-647-microtubules immobilised on the coverslip. Six hundred picomolar KIF1C-GFP was flown into a chamber with MRB80, 1 mM AMP-PNP, 100 μM taxol and 0.2 mg/ml κ-casein. Images were acquired every 500 ms for 600 s at an exposure of 400 ms using the 488 nm laser line at 50% laser power. Images were analysed using the Plot Profile function in ImageJ for each individual spot and manually counting bleach steps. A mixed binomial distribution for dimers and tetramers of KIF1C was fitted to the data using Eq. (1), where $P(k)$ is the probability to find $k$ bleach steps, $p$ is the fraction of active GFP and $x$ is the fraction of tetramers.

$$P(k) = \left[\frac{2!}{k!(2-k)!} \times p^k(1-p)^{2-k}\right] \times (1-x) + \left[\frac{4!}{k!(4-k)!} \times p^k(1-p)^{4-k}\right] \times (x).$$

(1)

**Statistical analysis and figure preparation.** Statistical data analyses and graphs were generated using Origin Pro 8.5 (OriginLab), python's Matplotlib and Scipy packages or R. Box plots show quartiles with whiskers indicating 10% and 90% of data as well as the mean indicated with a circle. All statistical significance analyses were carried out using two-tailed two-sample t-tests assuming equal variance (referred to a t-test is legends), Mann–Whitney U-test, ANOVA with post hoc Tukey, or Kolmogorov–Smirnov test. Where necessary, p-values were adjusted for multiple comparisons using Bonferroni or Holm–Šídák corrections. Figures were prepared by adjusting min/max and applying and inverting look up tables using ImageJ and assembled using Adobe Illustrator.

**Reporting summary.** Further information on research design is available in the Nature Research Reporting Summary linked to this article.

## Data availability
Data supporting the findings of this manuscript are available from the corresponding author upon reasonable request. A reporting summary for this Article is available as a Supplementary Information file. The source data underlying Figs. 1a, 1e, 3c, 3d, 3f, 3g, 3h, 4c, 4d, 4f, 5d, 5e, 6c, 6j, 6k and 7c, 7f and Supplementary Figs 2e and 6 are provided as a Source Data file. Our BioID mass spectrometry data are provided in Supplementary Data 1. The crosslinking mass spectrometry data have been deposited to the ProteomeXchange Consortium via the PRIDE[59] partner repository with the dataset identifier PXD013939.

## Code availability
Custom ImageJ macros will be available at the CMCB Github (https://github.com/cmcb-warwick) and on our website (http://mechanochemistry.org/Straube/#tab=soft) upon publication.

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

## Acknowledgements

We thank Antonio Feliciello for PTPN21 plasmids, Alex Jones and the WPH Proteomics RTP at the University of Warwick for mass spectrometry, the BBSRC and EPSRC-funded Warwick Integrative Synthetic Biology (WISB) centre Research Technology Facility for access to microscale thermophoresis equipment, Robert Dallmann for providing mouse brains, Steve Royle for help with hippocampus dissection, and Masanori Mishima for the mixed binomial model and comments on the manuscript.

This work was funded by a Research Prize from the Lister Institute of Preventive Medicine, a Wellcome Trust Investigator (200870/Z/16/Z) and a non-clinical Ph.D. studentship from the British Heart Foundation (FS/13/42/30377) to A.S., a Lister Institute Summer Studentship and a Chancellor's International PhD Scholarship from the University of Warwick that supported N.S., and a Ph.D. studentship from the EPSRC-funded MOAC CTD (EP/F500378/1) that supported H.H. A.J.Z. is funded by the MRC Doctoral Training Partnership (MR/N014294/1). D.R. is supported in part from the University of Warwick. J.B. is funded by a WMS Scholarship from the University of Warwick.

## Author contributions

A.S. and I.K. perceived the project, N.S. purified KIF1C, KIF1CΔCC3 and FERM domains, performed biochemical characterisation, single molecule experiments, crosslink mass spectrometry, KIF1C localisation in cells and data analysis. A.J.Z. performed BioID, purified Hook3, KIF1CΔS, KIF1C motor and stalk constructs, performed single-molecule experiments and data analysis, A.B. performed all cell biology experiments except the KIF1C/KIF16B double depletion performed by J.B., DR generated resources, performed coIPs and image analysis, H.H. analysed crosslink mass spectrometry data, A.S. managed the project, analysed data and wrote the manuscript with contributions from all authors.

## Additional information

**Competing interests:** The authors declare no competing interests.

