## [Peer Review File · Nature Communications]

Reviewers' Comments:

Reviewer #1:

Remarks to the Author:

In this study, Siddiqui and colleagues examine the molecular basis for the regulation of the kinesin-3 family member KIF1C, a motor protein that has been previously implicated in a range of intracellular transport processes. Analysis of KIF1C expressed in and purified from insect cells, using a range of in vitro techniques, indicates that in contrast to other kinesin-3 family members, KIF1C exists as a stable dimer that can undergo a conformational change from a relatively compact conformation to a more elongated conformation, that is promoted by an increase in salt concentration. To better understand the nature of the more compact conformation, they perform cross-linking mass-spectrometry analysis which highlights a potential intramolecular interaction between the amino terminal motor domain's microtubule binding interface and domains/regions toward the middle of the protein containing a coiled-coil and a FHA domain. They suggest that such an interaction would likely inhibit the activity of KIF1C and so propose this as a means of regulation (as opposed to a monomer-dimer transition that is generally believed to be the main mode of regulation of other kinesin-3 family members).

Previous work has indicated that KIF1C is important for podosome formation and they show that a previously identified KIF1C binding partner, PTPN21, is also important for this, and that expression of PTPN21 can compensate for partial loss of KIF1C in podosome formation, integrin vesicle trafficking and the transport of dense core-granules, proposing that PTPN21 activates the remaining pool of KIF1C. This is supported by data showing that PTPN21 increases the landing rate and frequency of running motors in an in vitro system. Further analysis indicates that PTPN21 alters the cross-linking mass spec profile of KIF1C, consistent with relief of autoinhibition. Deletion of the CC3 region of KIF1C implicated in forming intramolecular interactions also relieves autoinhibition in vitro and this is not further enhanced by PTPN21. Together, these data suggest that PTPN21 can activate KIF1C.

Finally, to determine whether this might be a general KIF1C activation mechanism, the authors perform Bio-ID using WT and CC3 deletion constructs and identify Hook3 (a previously defined KIF1C binding partner) as a protein that binds to WT but not the CC3 deletion. As for PTPN21, Hook3 appears to have an activating effect on KIF1C in vitro.

Overall, the data are of high quality. This mode of regulation of KIF1C by PTPN21 is a novel finding and is important given the apparently different mode of regulation for other kinesin-3 family members. The extension of the findings to Hook3 suggest that it may be of broader importance. Some concerns are outlined below that I expect the authors can address.

Statistical analysis is performed well and the methods provide sufficient detail.

Specific comments:

1. I am not fully convinced by the author's interpretation of the KIF1C partial knockdown/PTPN21 rescue data. They conclude that the reason why PTPN21 can rescue KIF1C knockdown is that the remaining pool is made more active. However, it also seems equally possible that an alternative pathway is activated by PTPN21 that is KIF1C independent. This could be addressed by efforts to more completely suppress KIF1C expression using siRNA or disruption of the gene using CRISPR-Cas9. If the author's proposed mechanism is correct, there should be no rescue if there is little/no KIF1C.

2. If this reviewer's interpretation of the author's data is correct, they did not identify any intra-dimer cross-links for the proposed stable KIF1C dimer, under conditions why they did identify intramolecular contacts. This seems very surprising and would seem to contradict the model. This seems particularly important given that PTPN21 did promote the formation of an intradimer cross-

link and so might suggest a 'cargo-induced' dimerization mechanism more akin to other kinesin-3 family members. This there an explanation for this?

3. Since receiving this manuscript to review, I noted that another paper has appeared on a pre-print server (Kendrick et al. <https://doi.org/10.1101/508887>) exploring KIF1C regulation and Hook3. Some of those findings are apparently contradictory to those presented here, most notably that purified KIF1C (from mammalian cells) is active and that activity is NOT affected by Hook3. I don't believe that this should affect consideration of this manuscript, and that this paper should stand on its own merits, but I would encourage the authors to comment on these other findings in their discussion in any revision.

4. It would be appropriate to show biochemically that deletion of CC3 does inhibit binding to PTPN21 and Hook3.

Reviewer #2:

Remarks to the Author:

This manuscript explores regulatory mechanisms of KIF1C and reports several novel findings. First of all, the authors demonstrate that purified KIF1C is dimeric and appears to undergo a large scale conformational change upon increase ionic strength. This suggests that the motor can fold back onto itself, which could reflect an autoinhibited state that can be relieved upon binding of a regulatory protein or cargo to the tail of this motor.

The authors propose that PTPN21, a protein known to interact with the KIF1C tail, is such a regulatory protein, given that depletion of this protein in smooth muscle cells phenocopies KIF1C depletion (i.e. reduction in podosomes). Moreover, overexpression of (fragments of) PTPN21 rescues podosome loss observed upon (partial) depletion of KIF1C. Furthermore, the authors report that such overexpression can also rescue transport defects cause by KIF1C depletion in RPE cells and hippocampal neurons, and take this as evidence that PTPN21 can activate KIF1C dependent transport.

Next, the authors probe how PTPN21 affect KIF1C motility in in vitro assays and report a 40% increase in the landing rate of this motor, which also display quite nice motility in the absence of this protein. Then, the authors start searching for other KIF1C interactors and identify Hook3 as a novel interaction partner of the KIF1C tail. This is interesting, given that Hook3 is a known activator for dynein/dynactin and that Hook3 can be an adaptor for both motor and perhaps coordinate their motility. The authors demonstrate that addition of Hook3 to motility assays result in a 100% increase of the landing rate, consistent with a regulatory function.

Overall, these are interesting result of sufficient importance and novelty to warrant publication in Nature Communications. The regulation of motors by adaptor proteins is of great current interest and the identification of a protein that can activate both dynein and a kinesin is quite exciting (and rather puzzling). This is reflected by a competing publication of another lab currently posted on BioRxiv (<https://www.biorxiv.org/content/early/2018/12/31/508887>), which also reports the KIF1C-Hook3 interaction (but not the activation of KIF1C). However, I do have a number of comments and reservations that need to be addressed.

1. Autoinhibition and activation

The authors interpret their data in the framework of autoinhibition and activation, i.e. the assume the protein folds back onto itself, thereby losing the ability to bind microtubules. Binding of a regulatory protein to tail domain competes with this interaction and results in activation. While this is a classical framework to think about KIF5, the current data does not necessarily warrant translation to KIF1C and there seem to be several instances of overinterpretation.

For example, the title of Figure 1 states that KIF1C is an autoinhibited dimer, but the result do no show that. They shows evidence for a strong conformational change upon increased ionic strength

and some evidence of interactions between different domains based on crosslinking mass spectrometry. There is, however, no evidence for autoinhibition in this figure.

In addition, while overexpression of PTPN21 nicely rescues the podosome loss upon KIF1C depletion, its effects on KIF1C motility *in vitro* are quite mild (40% more landing, compared to 200% more landing upon truncation of the tail). If this protein would really induce a switch for inhibition to activation, a more dramatic change would be expected (including less motility with KIF1C). Obviously, this all depends on the exact buffer conditions (e.g. ionic strength) and relative protein concentration, but as it stands, the PTPN21 thread is a bit thin. For example, podosome rescue could also be caused by overactivation of another motor (as the authors also discuss, but not exclude experimentally). For the cellular experiments, it could be tested if PTPN21 also rescues podosome formation and transport when KIF1C and KIF16B are both depleted. For the *in vitro* assay, titration of ionic strength and relative protein concentrations could be explored.

In addition, the effects of PTPN21 on cellular transport are not so clear. First of all, the kymographs in Figure 3e do not appear very striking. For figure 3f, it would be good to have a reference indicating which direction is retrograde/anterograde. Moreover, please clarify if this is data from axons or dendrites. Finally, the analysis should be split between anterograde and retrograde transport. At first glance, there seems to be a striking increase in movement to the right upon pFERM overexpression.

2. Hook3 results and overall flow of the manuscript

As indicated above, I find the Hook3 result very interesting. While it also seems to be a more potent 'activator', most of the figures concerns PTPN21 and only one panel covers Hook3 (Figure 5e/f). Here the authors show very nice co-motility, which is not shown for PTPN21. Surprisingly, no biochemical data on the identification of Hook3 as binding partner of KIF1C is shown.

Overall, this makes this manuscript a bit of an odd read. I think there are several ways to improve this and would encourage the authors to do so to strengthen their story. While it would be advisable to include more data, this should be balanced against the possible time pressure due to the competing manuscript in coordination with the editor.

Reviewer #3:

Remarks to the Author:

This is a very well written and comprehensive study addressing the regulation of the organelle transporter kinesin-3, KIF1C. The authors show very nicely that KIF1C is activated by phosphatase PTPN21. To investigate the mechanism of this inactivation, the authors conducted numerous *in vivo* studies using variants of KIF1C and PTPN21, focusing on the evaluation of the podosome formation in muscle, transfer of vesicles in neurons and microtubules assembly *in vitro*.

Moreover, the authors conducted a biochemical analysis of the KIF1C protein to characterize the oligomeric form of the inactive form of KIF1C as well as define the interacting site of the KIF1C - PTPN21 complex. The authors claim that KIF1C is a dimer in the inactive form and that interaction of PTPN21 with the KIF1C's coiled-coil domain leads to the KIF1C activation. This is based mainly on the XL-MS analysis. Unfortunately, this part is not developed enough and additional pieces of evidence and discussions should be provided.

To sum up, the presented study is novel, has potentially high impact in the field. My main concerns focus on the biochemical analysis of the dimer form which is suggested to be important for inactivation of KIF1C as well as direct interaction with PTPN21 which is proposed to alter this dimer-dependent inactivation.

1. An oligomeric form of KIF1C

p. 3 lines 79- 83. The authors have used based the glycerol gradient and size exclusion analysis to

suggest that KIF1C is a dimer. The related figure does not fully support this statement. Based on the SDS-PAGE analysis size of the monomer (KIF1C-GFP) should be around 190 kDa. Glycerol gradient analysis suggests that the majority of the KIF1C-GFP protein elutes in the range of 120-240 kDa (depending on the NaCl concentration). Interestingly, high salt (500mM) leads to more compact form (in the range of the GOX dimer protein, 120kDa) than low salt (50-100mM, smaller than catalase trimer, ~240kDa).

Contrarily, size exclusion suggests that low salt leads to a more compact structure. Moreover, oligomeric state evaluation in size exclusion reveals that KIF1C-GFP is indeed found in its dimer (or even trimer state). The more precise analysis should be done. SEC-MALS or equivalent can provide more concrete evaluation of the oligomeric state, its dynamics and its salt dependence.

If the authors decide to use the current analysis, additional data should be provided such as SDS-PAGE analysis of the standard proteins in the glycerol gradient analysis. Most probably elution profile of the standard proteins also depends on salt concentration.

Size exclusion profiles of the standard proteins.

In addition, the authors used a fusion protein with GFP, which might interfere stability of the oligomeric structure of KIF1C.

The main point of the authors that PTPN21 binds to KIF1C and disturb the oligomeric state. This should be shown by either size exclusion or SEC-MALS or other methods.

2. The XL-MS analysis of KIF1C.

The authors made a lot of efforts to show that KIF1C is dimer and maybe a trimer. Then, by applying the BS3 and EDC mediated crosslinking, the authors suggest few potential inter-domain interactions within KIF1C

Since KIF1C is a dimer, it is difficult to say if these are inter and intra interactions. Moreover, based on the method session, the authors used mM protein concentration, which most probably results in trimer and higher oligomeric species, may be mixed population of different forms. This should be evaluated by SDS-PAGE. Nevertheless, based on the XL-MS analysis, it is unclear if the identified binding sites are within a single monomer or between two monomers.

In the method session: "A cross-linked peptide was considered as valid if it achieved a StavroX score of at least 100, ..." what was the FDR used in this analysis? Did the crosslinks passed the FDR cutoff? Did crosslinks mentioned in Fig 1 and in the Fig S1 were detected with a reasonable score in more than one replicate?

3. The interaction between KIF1C and PTPN21.

The authors have shown very nicely that PTPN21 affect the KIF1C inhibition. It is clear that the FERM domain is involved in this interplay. Moreover, based on Fig 3, most probably CC3 domain of KIF1C is required for this crosstalk. However, strong and direct evidence for binding between PTPN21 and KIF1C is lacking. The crosslinked fragments are of a very low intensity, less than 5% (peptides HKFYR and SGNR..) and poor fragmentation (peptides LEMEKR and HPVVFR..). The MS/MS spectrum of the HPVVFR.. is very partial, covers only a few amino acids in the C-terminal area of the peptide, which might suggest that this is a false positive crosslink.

Lack of good crosslinks doesn't mean that these proteins do not interact, however, additional pieces of evidence should be provided.

4. The Hook3 – KIF1C interactome.

The authors utilized the BioD2 analysis to investigate KIF1C interacting proteins, which specifically interact with the 623-825 region of KIF1C, which is also involved in the PTPN21 mediated activation. In this analysis, Hook3 protein was identified as a binder.

The authors should provide results of this proteomic analysis, which identified potential 240 proteins. The community can benefit from this analysis.

Moreover, all the required statistical analysis should be provided, as well as, I strongly suggest to

upload the proteomic data to a public server.

Minor comments:

Fig 1b-c: Size of the standard proteins should be included in the figure.

Fig 1d is related to Fig 1b, therefore it is better to place them together.

Fig 1g- specify the EDC and the BS3 mediated crosslinks.

Fig 1 doesn't address the activity or inhibition of the KIF1C dimer. Therefore, the title should be reconsidered.

Fig 2. (legend). "Statistical significance with $p > 0.05$ is indicated with asterisks", should be < 0.05 .

Fig 2g-h. To be consistent, please use either PTPN21 or PTPD1

We thank all three reviewers for their constructive criticism. We have marked changes in the manuscript text by yellow highlighting and provide a detailed response to all comments below.

Reviewer comments:

Reviewer #1 (Remarks to the Author):

In this study, Siddiqui and colleagues examine the molecular basis for the regulation of the kinesin-3 family member KIF1C, a motor protein that has been previously implicated in a range of intracellular transport processes. Analysis of KIF1C expressed in and purified from insect cells, using a range of in vitro techniques, indicates that in contrast to other kinesin-3 family members, KIF1C exists as a stable dimer that can undergo a conformational change from a relatively compact conformation to a more elongated conformation, that is promoted by an increase in salt concentration. To better understand the nature of the more compact conformation, they perform cross-linking mass-spectrometry analysis which highlights a potential intramolecular interaction between the amino terminal motor domain's microtubule binding interface and domains/regions toward the middle of the protein containing a coiled-coil and a FHA domain. They suggest that such an interaction would likely inhibit

the activity of KIF1C and so propose this as a means of regulation (as opposed to a monomer-dimer transition that is generally believed to be the main mode of regulation of other kinesin-3 family members).

Previous work has indicated that KIF1C is important for podosome formation and they show that a previously identified KIF1C binding partner, PTPN21, is also important for this, and that expression of PTPN21 can compensate for partial loss of KIF1C in podosome formation, integrin vesicle trafficking and the transport of dense core-granules, proposing that PTPN21 activates the remaining pool of KIF1C. This is supported by data showing that PTPN21 increases the landing rate and frequency of running motors in an in vitro system. Further analysis indicates that PTPN21 alters the cross-linking mass spec profile of KIF1C, consistent with relief of autoinhibition. Deletion of the CC3 region of KIF1C implicated in forming intramolecular interactions also relieves autoinhibition in vitro and this is not further enhanced by PTPN21. Together, these data suggest that PTPN21 can activate KIF1C.

Finally, to determine whether this might be a general KIF1C activation mechanism, the authors perform Bio-ID using WT and CC3 deletion constructs and identify Hook3 (a previously defined KIF1C binding partner) as a protein that binds to WT but not the CC3 deletion. As for PTPN21, Hook3 appears to have an activating effect on KIF1C in vitro.

Overall, the data are of high quality. This mode of regulation of KIF1C by PTPN21 is a novel finding and is important given the apparently different mode of regulation for other kinesin-3 family members. The extension of the findings to Hook3 suggest that it may be of broader importance. Some concerns are outlined below that I expect the authors can address.

Statistical analysis is performed well and the methods provide sufficient detail.

Specific comments:

1. I am not fully convinced by the author's interpretation of the KIF1C partial knockdown/PTPN21 rescue data. They conclude that the reason why PTPN21 can rescue KIF1C knockdown is that the remaining pool is made more active. However, it also seems equally possible that an alternative pathway is activated by PTPN21 that is KIF1C independent. This could be addressed by efforts to more completely suppress KIF1C expression using siRNA or disruption of the gene using CRISPR-Cas9. If the author's proposed mechanism correct, there should be no rescue if there is little/no KIF1C.

We agree with the reviewer and performed additional experiments to formally test this idea. Based on the finding that the FERM domain of PTPN21 also interacts with KIF16B (Carlucci *et al.*, 2010), we decided to follow up the alternative hypothesis that suppression of the KIF1C phenotype might be due to PTPN21 activating KIF16B to compensate for KIF1C depletion. New data that we show in Figure 4d and S4 show a mild podosome formation phenotype for KIF16B depletion. However, FERM domain expression increased the number of podosomes in control cells and when either KIF1C or KIF16B were depleted, but not when both KIF1C and KIF16B were depleted simultaneously. These data are strong evidence that FERM domain suppresses KIF1C phenotypes by activating alternative kinesins rather than the remaining pool of KIF1C. We therefore removed this statement in the revised manuscript.

2. If this reviewer's interpretation of the author's data is correct, they did not identify any intra-dimer cross-links for the proposed stable KIF1C dimer, under conditions why they did identify intramolecular contacts. This seems very surprising and would seem to contradict the model. This seems particularly important given that PTPN21 did promote the formation of an intradimer cross-link and so might suggest a 'cargo-induced' dimerization mechanism more akin to other kinesin-3 family members. This there an explanation for this?

To address this concern, we have carefully reassessed our crosslinking mass spectrometry experiments and performed additional repeats, including experiments under high salt conditions. We find a very low abundance of the reported intradimer crosslink and there is a technical limitation to identify crosslinks in parallel protein stands as any crosslinks to neighbouring peptides are indistinguishable from partially cleaved peptides that have been modified by a crosslinker molecule (dead-end). We therefore reject all those potential crosslinks. Using the a narrow search window of peptide mass and retention time for the intradimer crosslink that we identified within the coiled-coil 3 domain, we do find comparable low amounts of the precursor ion in KIF1C alone both at 150mM salt, 500mM and in the presence of FERM domain. We have therefore added this intradimer crosslink to the KIF1C data shown in Figures 2a and 5f.

Furthermore, our single-molecule data do not support a cargo-induced dimerization mechanism. KIF1C-GFP moves processively on microtubules in the absence of PTPN21 or a cargo adaptor, and our photobleaching analysis shows a dimeric protein. Together with the hydrodynamic analysis of both purified KIF1C and KIF1C in cell lysates, we are confident that KIF1C forms a stable dimer.

3. Since receiving this manuscript to review, I noted that another paper has appeared on a pre-print server (Kendrick *et al.* <https://doi.org/10.1101/508887>) exploring KIF1C regulation and Hook3. Some of those findings are apparently contradictory to those presented here, most notably that purified KIF1C (from mammalian cells) is active and that activity is NOT affected by Hook3. I don't believe that this should affect consideration of this manuscript, and that this paper should stand on its own merits, but I would encourage the authors to comment on these other findings in their discussion in any revision.

We have included a reference to the mentioned preprint in the discussion as it provides direct evidence for the complex of dynein/dynactin, Hook3 and KIF1C. We also discuss the fact that Kendrick et al. didn't find any evidence for Hook3 activating KIF1C. However, they do only report speed and run lengths of motors, not their landing rates. We do not see a significant increase in speed and run length with PTPN21 or Hook3 either. Due to the interaction of the stalk of KIF1C blocking the microtubule binding site, the activators of KIF1C predominantly increase the landing rate. Thus the study by Kendrick et al. is consistent with our findings and maps the Hook3 interaction site to the same region in the stalk. Having said this, we now included new data that show that an increase in salt concentration is sufficient to open KIF1C and remove the stalk-motor contacts (Figure S1e), so it is possible that different posttranslational modifications or buffer conditions might result in a constitutively activated motor preparation.

4. It would be appropriate to show biochemically that deletion of CC3 does inhibit binding to PTPN21 and Hook3.

While the deletion of CC3 significantly increases the landing rate, probably by reducing the affinity of stalk and motor domains, this does not mean that CC3 is essential to bind the activators. However, if we remove a larger region of the stalk, then we further increase the landing rate of KIF1C motors and also remove the interaction sites for PTPN21 and Hook3. We now show both using co-IPs (Figures 7C and S6) and TIRF assays (Figure 8a-d) that binding of PTPN21 and Hook3 dramatically reduced binding to KIF1C when we delete amino acids 612-922.

Reviewer #2 (Remarks to the Author):

This manuscript explores regulatory mechanisms of KIF1C and reports several novel findings. First of all, the authors demonstrate that purified KIF1C is dimeric and appears to undergo a large scale conformational change upon increase ionic strength. This suggests that the motor can fold back onto itself, which could reflect an autoinhibited state that can be relieved upon binding of a regulatory protein or cargo to the tail of this motor. The authors propose that PTPN21, a protein known to interact with the KIF1C tail, is such a regulatory protein, given that depletion of this protein in smooth muscle cells phenocopies KIF1C depletion (i.e. reduction in podosomes). Moreover, overexpression of (fragments of) PTPN21 rescues podosome loss observed upon (partial) depletion of KIF1C. Furthermore, the authors report that such overexpression can also rescue transport defects cause by KIF1C depletion in RPE cells and hippocampal neurons, and take this as evidence that PTPN21 can activate KIF1C dependent transport. Next, the authors probe how PTPN21 affect KIF1C motility in in vitro assays and report a 40% increase in the landing rate of this motor, which also display quite nice motility in the absence of this protein. Then, the authors start searching for other KIF1C interactors and identify Hook3 as a novel interaction partner of the KIF1C tail. This is interesting, given that Hook3 is a known activator for dynein/dynactin and that Hook3 can be an adaptor for both motor and perhaps coordinate their motility. The authors demonstrate that addition of Hook3 to motility assays result in a 100% increase of the landing rate, consistent with a regulatory function. Overall, these are interesting result of sufficient importance and novelty to warrant publication in Nature Communications. The regulation of motors by adaptor proteins is of great current interest and the identification of a protein that can activate both dynein and a kinesin is quite exciting (and rather

puzzling). This is reflected by a competing publication of another lab currently posted on BioRxiv (<https://www.biorxiv.org/content/early/2018/12/31/508887>), which also reports the KIF1C-Hook3 interaction (but not the activation of KIF1C). However, I do have a number of comments and reservations that need to be addressed.

1. Autoinhibition and activation

The authors interpret their data in the framework of autoinhibition and activation, i.e. they assume the protein folds back onto itself, thereby losing the ability to bind microtubules. Binding of a regulatory protein to tail domain competes with this interaction and results in activation. While this is a classical framework to think about KIF5, the current data does not necessarily warrant translation to KIF1C and there seem to be several instances of over interpretation.

For example, the title of Figure 1 states that KIF1C is an autoinhibited dimer, but the results do not show that. They show evidence for a strong conformational change upon increased ionic strength and some evidence of interactions between different domains based on crosslinking mass spectrometry. There is, however, no evidence for autoinhibition in this figure.

We agree with these comments and did two additional experiments to directly address the autoinhibition and summarise the data in a large new Figure 2. We purified the motor and stalk domains of KIF1C separately and show that both bind to each other with an affinity of 1 μ M. We also did microtubule binding experiments with the motor domain construct and show that adding 1 μ M of KIF1C stalk reduces microtubule binding significantly compared to adding 1 μ M GST as control. We believe these experiments provide direct evidence for our proposed autoinhibition mechanism.

In addition, while overexpression of PTPN21 nicely rescues the podosome loss upon KIF1C depletion, its effects on KIF1C motility *in vitro* are quite mild (40% more landing, compared to 200% more landing upon truncation of the tail). If this protein would really induce a switch for inhibition to activation, a more dramatic change would be expected (including less motility with KIF1C). Obviously, this all depends on the exact buffer conditions (e.g. ionic strength) and relative protein concentration, but as it stands, the PTPN21 thread is a bit thin. For example, podosome rescue could also be caused by overactivation of another motor (as the authors also discuss, but not exclude experimentally). For the cellular experiments, it could be tested if PTPN21 also rescues podosome formation and transport when KIF1C and KIF16B are both depleted. For the *in vitro* assay, titration of ionic strength and relative protein concentrations could be explored.

We did the suggested experiment and show that PTPN21 FERM increases podosome numbers unless both KIF1C and KIF16B are depleted. These new data are in Figure 4d and S4. The new data suggest that PTPN21 activates both KIF1C and KIF16B. We now include data with a larger stalk deletion, which has an even more dramatic effect on KIF1C activity, which supports the importance of the stalk region for autoinhibition. This region is required for efficient interaction with PTPN21 (new coIP in Figure S6 and new TIRF data in Figure 8). With regards to the magnitude of the effect observed *in vitro*, one needs to consider that an important technical limitation of single molecule *in vitro* assays is that these have to be performed at very low protein concentrations. Therefore, observing effects that depend on micromolar affinity interactions, is difficult. The concentration range that allows counting single molecule landing events is limited and likely to be well below physiological protein concentrations. Ionic strength is important too and we include new data that demonstrate that the motor-to-stalk crosslinks that are abundant in the presence of 150mM salt are largely absent at

500mM (Figure S2e). Mixing KIF1C and PTPN21 at 0.5 μ M each is even more efficient in removing the motor-stalk crosslinks (Figure 5e), but this concentration is not compatible with single molecule experiments. Therefore, we do not agree that being able to observe co-transport and a significant increase in landing rate at low nanomolar concentrations is weak evidence for PTPN21 as KIF1C activator.

In addition, the effects of PTPN21 on cellular transport are not so clear. First of all, the kymographs in Figure 3e do not appear very striking. For figure 3f, it would be good to have a reference indicating which direction is retrograde/anterograde. Moreover, please clarify if this is data from axons or dendrites. Finally, the analysis should be split between anterograde and retrograde transport. At first glance, there seems to be a striking increase in movement to the right upon pFERM overexpression.

We agree that our choice of example kymograph for siKIF1C did not represent the quantitative data well and have replaced this – this is now Figure 4e. All our kymograph examples are now labelled to show that cell body / minus ends are consistently at the left. While we did not label our neurites to distinguish dendrites and axons, we did separate anterograde/retrograde in our analysis as suggested. This confirmed the impression of the reviewer that the increase in transport activity is predominantly in anterograde direction – consistent with kinesin activation. These data are in Figure 4h and 4i.

2. Hook3 results and overall flow of the manuscript

As indicated above, I find the Hook3 result very interesting. While it also seems to be a more potent 'activator', most of the figures concerns PTPN21 and only one panel covers Hook3 (Figure 5e/f). Here the authors show very nice co-motility, which is not shown for PTPN21. Surprisingly, no biochemical data on the identification of Hook3 as binding partner of KIF1C is shown.

Overall, this makes this manuscript a bit of an odd read. I think there are several ways to improve this and would encourage the authors to do so to strengthen their story. While it would be advisable to include more data, this should be balanced against the possible time pressure due to the competing manuscript in coordination with the editor.

We repeated the BioID experiments to be able to provide solid quantitative analysis of the data and show these in a volcano plot in Figure 7b and a supplementary table. We also performed co-immunoprecipitation to demonstrate biochemical evidence for stalk-dependent binding of Hook3 to KIF1C, shown in new Figure 7c. Furthermore, we show co-motility for both Hook3 and PTPN21 FERM in Figure 8 and that this depends on the stalk domain of KIF1C.

Reviewer #3 (Remarks to the Author):

This is a very well written and comprehensive study addressing the regulation of the organelle transporter kinesin-3, KIF1C. The authors show very nicely that KIF1C is activated by phosphatase PTPN21. To investigate the mechanism of this inactivation, the authors conducted numerous in-vivo studies using variants of KIF1C and PTPN21, focusing on the evaluation of the podosome formation in muscle, transfer of vesicles in neurons and microtubules assembly in vitro.

Moreover, the authors conducted a biochemical analysis of the KIF1C protein to characterize the oligomeric form of the inactive form of KIF1C as well as define the interacting site of the KIF1C-

PTPN21 complex. The authors claim that KIF1C is a dimer in the inactive form and that interaction of PTPN21 with the KIF1C's coiled-coil domain leads to the KIF1C activation. This is based mainly on the XL-MS analysis. Unfortunately, this part is not developed enough and additional pieces of evidence and discussions should be provided.

To sum up, the presented study is novel, has potentially high impact in the field. My main concerns focus on the biochemical analysis of the dimer form which is suggested to be important for inactivation of KIF1C as well as direct interaction with PTPN21 which is proposed to alter this dimer-dependent inactivation.

1. An oligomeric form of KIF1C

p. 3 lines 79- 83. The authors have used based the glycerol gradient and size exclusion analysis to suggest that KIF1C is a dimer. The related figure does not fully support this statement. Based on the SDS-PAGE analysis size of the monomer (KIF1C-GFP) should be around 190 kDa. Glycerol gradient analysis suggests that the majority of the KIF1C-GFP protein elutes in the range of 120-240 kDa (depending on the NaCl concentration). Interestingly, high salt (500mM) leads to more compact form (in the range of the GOX dimer protein, 120kDa) than low salt (50-100mM, smaller than catalase trimer, ~240kDa).

Contrarily, size exclusion suggests that low salt leads to a more compact structure. Moreover, oligomeric state evaluation in size exclusion reveals that KIF1C-GFP is indeed found in its dimer (or even trimer state). The more precise analysis should be done. SEC-MALS or equivalent can provide more concrete evaluation of the oligomeric state, its dynamics and its salt dependence.

If the authors decide to use the current analysis, additional data should be provided such as SDS-PAGE analysis of the standard proteins in the glycerol gradient analysis. Most probably elution profile of the standard proteins also depends on salt concentration. Size exclusion profiles of the standard proteins.

The molecular weight of a non-globular protein cannot be determined by glycerol gradient or size exclusion chromatography alone, but requires both to be analysed – or indeed use of alternative techniques such as analytical ultracentrifugation or scattering techniques such as SEC-MALS. As KIF1C is an elongated protein, it will behave like a smaller globular protein than its molecular weight in a glycerol gradient, but runs with larger proteins in gel filtration. The standard proteins are all more or less globular, but several of them fall apart at high salt. Thus we calibrated the glycerol gradients and gel filtration experiments at 150mM salt and label the position of standard proteins using the published sedimentation coefficients and Stoke's radii for the standard proteins. It would be misleading to give their molecular weights as this would only apply to globular proteins. The traces for standard proteins are shown in new Figure S1 as requested. Determining the Stoke's radius and sedimentation coefficient from SEC and glycerol gradients respectively, allowed us to calculate the apparent molecular weight for KIF1C, which is consistent with a dimer.

In addition, the authors used a fusion protein with GFP, which might interfere stability of the oligomeric structure of KIF1C.

eGFP dimerises with a K_d of 0.11 mM (Snapp et al J Cell Biol 2003). We are working well below of these concentrations, but as a sanity check, we run eGFP over gel filtration in the same buffer as KIF1C-GFP and GFP was a pure monomer under those conditions. To address the reviewer's concern more directly, we decided to undertake hydrodynamic analysis of KIF1C-Flag from cell extracts. While a significant part of the protein is engaged in large complexes that ended up in the

void volume of the gel filtration column, the other half of KIF1C behaved similar to the insect cell purified KIF1C dimers. These new data can be found in Figure S2.

The main point of the authors that PTPN21 binds to KIF1C and disturb the oligomeric state. This should be shown by either size exclusion or SEC-MALS or other methods.

We propose that KIF1C is a stable dimer that undergoes a conformational change upon activation (Figure 8e), which is contrary to the idea that cargo or adapter binding results in oligomerization of the motor. Binding of PTPN21 engages the stalk region that otherwise blocks the microtubule binding region of the motor domain (data in Figures 2, 5, 8 and S6). We believe our data support this model and that SEC of the KIF1C-PTPN21 complex is unlikely to reveal further insights into the activation mechanism.

2. The XL-MS analysis of KIF1C.

The authors made a lot of efforts to show that KIF1C is dimer and maybe a trimer. Then, by applying the BS3 and EDC mediated crosslinking, the authors suggest few potential inter-domain interactions within KIF1C. Since KIF1C is a dimer, it is difficult to say if these are inter and intra interactions. Moreover, based on the method session, the authors used mM protein concentration, which most probably results in trimer and higher oligomeric species, may be mixed population of different forms. This should be evaluated by SDS-PAGE. Nevertheless, based on the XL-MS analysis, it is unclear if the identified binding sites are within a single monomer or between two monomers.

We used 1 μ M rather than 1mM of KIF1C for crosslinking. We do not consider that this is a concentration that should result in excessive protein aggregation. We agree that a limitation of XL-MS is that we cannot distinguish whether the Lysine in the stalk bound to the Lysine in the motor domain of the same or a different KIF1C molecule within the dimer or even across two dimers. In any case, the XL-MS experiment provided evidence for an interaction of stalk and motor domains, which we have verified with further biochemical experiments: Using microscale thermophoresis we measured the affinity of separately purified motor and stalk domain as 1 μ M (new data in Figure 2f). We also show that stalk domain interferes with microtubule binding of the motor domain (new data in Figure 2g-h), providing direct evidence for our proposed autoinhibition mechanism, which is entirely consistent with the identified crosslinks.

In the method session: “A cross-linked peptide was considered as valid if it achieved a StavroX score of at least 100, ...” what was the FDR used in this analysis? Did the crosslinks passed the FDR cutoff? Did crosslinks mentioned in Fig 1 and in the Fig S1 were detected with a reasonable score in more than one replicate?

The FDR used in the analysis was 5%. We added this information to methods section on page 29. All crosslinked peptides shown in our manuscript passed the FDR cutoff and a StavroX score of at least 100 in at least one sample. Albeit in some samples, the StavroX score needed to pass the FDR was lower than 100. Once identified crosslinked peptides were then known by their mass and retention time to identify and quantify their abundance across all samples. We explain this in the methods section on page 29. We did identify the predominant motor-stalk crosslink LEMEKR-LKEGANINK in 4 independent experiments of KIF1C treated with BS3.

3. The interaction between KIF1C and PTPN21.

The authors have shown very nicely that PTPN21 affect the KIF1C inhibition. It is clear that the FERM domain is involved in this interplay. Moreover, based on Fig 3, most probably CC3 domain of KIF1C is required for this crosstalk. However, strong and direct evidence for binding between PTPN21 and KIF1C is lacking. The crosslinked fragments are of a very low intensity, less than 5% (peptides HKFYR and SGNR..) and poor fragmentation (peptides LEMEKR and HPVVFVR..). The MS/MS spectrum of the HPVVFVR.. is very partial, covers only a few amino acids in the C-terminal area of the peptide, which might suggest that this is a false positive crosslink. Lack of good crosslinks doesn't mean that these proteins do not interact, however, additional pieces of evidence should be provided.

We now provide additional evidence for the KIF1C – PTPN21 interaction by co-immunoprecipitation (new data in Figure S6) and dual-colour TIRF experiments (Figure 8a-b). These data also show that the interaction is dramatically reduced when the stalk region is deleted from KIF1C.

4. The Hook3 – KIF1C interactome.

The authors utilized the BioID2 analysis to investigate KIF1C interacting proteins, which specifically interact with the 623-825 region of KIF1C, which is also involved in the PTPN21 mediated activation. In this analysis, Hook3 protein was identified as a binder. The authors should provide results of this proteomic analysis, which identified potential 240 proteins. The community can benefit from this analysis. Moreover, all the required statistical analysis should be provided, as well as, I strongly suggest to upload the proteomic data to a public server.

For the benefit of the community, we repeated the BioID experiments to be able to perform statistical analysis and thus share a meaningful dataset. We added a volcano plot in Figure 7b and a supplementary table that contains the mass spectrometry data and statistical analysis.

Minor comments:

Fig 1b-c: Size of the standard proteins should be included in the figure.

These figures are now Figure 1a and b. For the reasons outlined above, the molecular weight of standard proteins is not informative as we are analyzing a non-globular protein and we use the two techniques to determine the sedimentation coefficient and the Stoke's radius to calculate the apparent molecular weight from these two value. Therefore, S and R_s of the standard proteins are given in the figures as these are the relevant quantities.

Fig 1d is related to Fig 1b, therefore it is better to place them together.

It is true that the frictional coefficient is calculated from the sedimentation coefficient and the theoretical molecular weight alone. Both panels are next to each other in the rearranged Figure 1, but placed in the order in which they are referred to in the text.

Fig 1g- specify the EDC and the BS3 mediated crosslinks.

We have separated the data so that Figure S2 contains BS3 crosslinks and Figure S3 details EDC crosslinks. We also have labelled the crosslinks in the cartoons in Figures S2, S3 and S5 to clearly indicate the crosslinker.

Fig 1 doesn't address the activity or inhibition of the KIF1C dimer. Therefore, the title should be reconsidered.

We agree and have now added new data that address autoinhibition directly and these data are shown in a largely new Figure 2. Figures 1 and S1 now only address the oligomeric state.

Fig 2. (legend). "Statistical significance with $p > 0.05$ is indicated with asterisks", should be < 0.05 .

Well spotted, we have corrected this in the legend of what is now Figure 3.

Fig 2g-h. To be consistent, please use either PTPN21 or PTPD1

We corrected these.

Reviewers' Comments:

Reviewer #1:

Remarks to the Author:

The authors have constructively addressed my comments and I am pleased to recommend this manuscript for publication.

Reviewer #2:

Remarks to the Author:

The authors have addressed most of my comments. The results are now much more complete, balanced and convincing.

Reviewer #3:

Remarks to the Author:

The authors have addressed all my comments. Additional figures and descriptions were included, providing more details and new experiments supporting the author's statements.